

# A new digital elevation model of Antarctica derived from ICESat-2

Xiaoyi Shen[1,2], Chang-Qing Ke[1,2], Yubin Fan[1,2], Lhakpa Drolma[3]

[1] School of Geography and Ocean Science, Nanjing University, Nanjing, 210023, China
[2] Jiangsu Provincial Key Laboratory of Geographic Information Science and Technology, Nanjing University, Nanjing, 210023, China
[3] Institute of Tibetan Plateau Atmospheric and Environmental Sciences, Tibet Meteorological Bureau, Lhasa, 850000, China

*Correspondence to*: Chang-Qing Ke (kecq@nju.edu.cn)

**Abstract.** Antarctic digital elevation models (DEMs) are essential for human fieldwork, ice motion tracking and the numerical modelling of ice sheet. In the past thirty years, several Antarctic DEMs derived from satellite data have been published. However, these DEMs either have coarse spatial resolutions or aggregate observation spanning over several years, which limit their further scientific applications. In this study, the new-generation satellite laser altimeter Ice, Cloud, And Land Elevation Satellite-2 (ICESat-2) is used to generate a new Antarctic DEM for both the ice sheet and ice shelves. Approximately $4.69 \times 10^9$ ICESat-2 measurement points from November 2018 to November 2019 are used to estimate surface elevations at resolutions of 500 m and 1 km based on a spatiotemporal fitting method. Approximately 74% of Antarctica is observed, and the remaining observation gaps are interpolated using the ordinary kriging method. The DEM is formed from the estimated elevations in 500 m and 1 km grid cells, and is finally posted at the resolution of 500 m. National Aeronautics and Space Administration Operation IceBridge (OIB) airborne data are used to evaluate the generated Antarctic DEM (hereafter called the ICESat-2 DEM) in individual Antarctic regions and surface types. Overall, a median bias of 0.03 m and a root-mean-square deviation of 15.64 m result from approximately $5.2 \times 10^6$ OIB measurement points. The accuracy and uncertainty of the ICESat-2 DEM vary in relation to the surface slope and roughness, and more reliable estimates are found in the flat ice sheet interior. The ICESat-2 DEM is superior to previous DEMs derived from satellite altimeters for both spatial resolution and elevation accuracy and comparable to those derived from stereo-photogrammetry and interferometry. Similar results are found when comparing to elevation measurements from kinematic GNSS (GPS and the Russian GLONASS) transects. The elevations of high accuracy and ability of annual update make the ICESat-2 DEM an addition to the existing Antarctic DEM groups, and it can be further used for other scientific applications. The generated ICESat-2 DEM (including the map of uncertainty) can be downloaded from National Tibetan Plateau Data Center, Institute of Tibetan Plateau Research, Chinese Academy of Sciences at https://data.tpdc.ac.cn/en/disallow/9427069c-117e-4ff8-96e0-4b18eb7782cb/ (Shen et al., 2021, DOI: 10.11888/Geogra.tpdc.271448).



## 1 Introduction

Knowledge of the detailed surface topography in Antarctica is essential for human fieldwork, ice motion tracking (Bamber et al., 2000) and the numerical modelling of ice sheet (Cornford et al., 2015). Digital elevation models (DEMs) of Antarctica, for example, can be used for presenting the topography of ice sheets and ice shelves and thus provide a crucial reference for ice dynamics and glacier velocities (Wesche et al., 2007; Slater et al., 2018), which is necessary for Antarctic mass balance monitoring and potential sea level rise contribution estimation (Ritz et al., 2015; Mengel et al., 2018).

Due to the remoteness of Antarctica, most of the previously published Antarctic DEMs were derived from satellite or airborne data, e.g., elevation measurements from radar altimeters (Fricker et al., 2000; Helm et al., 2014; Slater et al., 2018), laser altimeters (DiMarzio et al., 2007), a combination of radar and laser altimeters (Bamber et al., 2009), stereo-photogrammetry (Korona et al., 2009; Cook et al., 2012; Howat et al., 2019) and interferometry (Wessel et al., 2021). The currently available continent-scale Antarctic DEMs include one DEM derived from ICESat (hereafter called the ICESat

DEM, DiMarzio et al., 2007), one based on the combination of ICESat and ERS-1 elevation measurements (hereafter called the ICESat/ERS-1 DEM, Bamber et al., 2009), two DEMs derived from CryoSat-2 (hereafter called the Helm CryoSat-2 DEM (Helm et al., 2014) and Slater CryoSat-2 DEM (Slater et al., 2018)), one DEM derived from stereo-photogrammetry using GeoEye-1 and WorldView-1/2/3 imageries (hereafter called the Reference Elevation Model of Antarctica (REMA) DEM, Howat et al., 2019), and one DEM derived from Interferometric Synthetic Aperture Radar (InSAR) using TerraSAR-

X and TanDEM-X data (hereafter called the TanDEM-X PolarDEM, Wessel et al., 2021).

All these DEMs provide reasonable elevation estimates for Antarctica; however, some flaws still cannot be totally avoided. The coverage of ICESat is limited in ice sheet margins due to its coarse across-track resolution, hence for ICESat DEM most of the elevations in ice sheet margins were interpolated based on the neighbour data. Although the ICESat/ERS-1 DEM improves the data coverage by combining the measurements from ICESat and ERS-1 elevations, this DEM aggregates

observation spanning over several years due to the different timespans (1994-1995 for ERS-1 and 2003-2008 for ICESat) of these two satellite altimeter datasets. This issue also exists with the REMA DEM and TanDEM-X PolarDEM, where multiyear satellite imageries were used. Different from the abovementioned DEMs, the Slater CryoSat-2 DEM was derived based on a model fitting method by using seven-year CryoSat-2 data (from July 2010 to July 2016). This method can quantify the measured elevation fluctuations due to interannual variations, and can provide a DEM for each month during the

timespan of applied data. However, although the radar penetration depth of the CryoSat-2 Ku-band into snowpack can be corrected for either empirically or theoretically using a waveform fitting approach (Davis, 1996; Davis, 1997), the spatial and temporal variations of radar penetration depth are still difficult to account. As multi-temporal and large-scale satellite radar altimeter data are usually used, the accuracy of estimated elevations is reduced. A similar problem also exists with the Helm CryoSat-2 DEM and TanDEM-X PolarDEM (the penetration depth of the X-band into snow may be several meters,

Fischer et al., 2020; Dehecq et al., 2016).



The new-generation satellite laser altimeter Ice, Cloud, And Land Elevation Satellite-2 (ICESat-2) of the National Aeronautics and Space Administration (NASA), which was launched on 15 September 2018, provides near-global (up to 88°S) and dense land ice elevation measurements in an accurate repeated cycle of 91 days by using a multibeam (six beams in three pairs that work at 532 nm) laser altimeter (i.e., Advanced Topographic Laser Altimeter System, ATLAS, Neumann

et al., 2019). The narrow footprint (approximately 17 m with a spatial interval of 0.7 m) and three pairs of beams (two beams in one pair can determine the local slope) enable a fine-scale measurement of Antarctic surface heights even in steep regions. Hence, ICESat-2 can be expected to provide a new Antarctic DEM on a fine scale.

Here, we use a one-year time series (from November 2018 to November 2019) of ICESat-2 elevation measurements to generate a new Antarctic DEM that covers both the ice sheet and ice shelves (hereafter called the ICESat-2 DEM). The

applied data, DEM generation method and quality control criteria are presented in Section 2. Furthermore, we present the map of the ICESat-2 DEM and construct an accuracy evaluation by comparing it to the elevation measurements from the NASA Operation IceBridge (OIB) airborne mission and kinematic GPS and the Russian GLONASS (GNSS) transects in Section 3. The performances of the ICESat-2 DEM and six currently available Antarctic DEMs are compared in Section 4, Section 5 provides the data availability and Section 6 concludes this study.

## 2 Data and Methods

### 2.1 ICESat-2 data

The ICESat-2 ATL06 land ice elevation product (Version 3, Smith et al., 2019) from November 2018 to November 2019 is used. This product provides land ice elevation measurements at a spatial resolution of 20 m after correcting instrument-specific biases (i.e., corrections for transmit-pulse shape and first-photon bias, Neumann et al. 2019); here, only ATL06 data

with good quality (according to the surface signal confidence metric from ATL06 data, i.e., those for which atl06_quality_summary equals zero) are used to generate the DEM. For the data collected over Antarctic ice shelves, corrections for ocean tide and inverse barometer effects are also applied (Egbert et al., 1994; Egbert and Erofeeva, 2002; Padman et al., 2002). Elevation measurements from all six beams are used to produce the densest surface height coverage. Although the signal energies of strong and weak beams are different, all six beams provide centimeter-scale elevation

measurements, and the biases of two beams in one pair are less than 2 cm (Brunt et al., 2019) and 5 cm (Shen et al., 2021) for flat and steep surfaces. Thus, the effect of elevations estimated from weak beams is negligible.

### 2.2 NASA OIB airborne data and kinematic GNSS data

Elevation measurements from the OIB airborne mission in Antarctica are used here to evaluate the accuracy of the ICESat-2 DEM on a continental scale, including in the stable ice sheet interior and active marginal ice shelves. Surface heights from

OIB airborne missions are measured by the Airborne Topographic Mapper (ATM), a conically scanning laser altimeter (at 532 nm) with a swath width of 140 m and footprint size of 1 to 3 m. The elevation measurement accuracy of ATM is



approximately 10 cm or better (Kurtz et al. 2013). Here, the IceBridge ATM L2 Icessn elevation, slope and roughness (V002) product (Studinger et al., 2014) is used, and a data filter (Young et al., 2008; Kwok et al., 2012; Studinger et al., 2014) is applied to remove abnormal values due to geolocation errors or cloud cover. The local terrain parameters, i.e., slope and

roughness, are calculated following Shen et al. (2021). To reduce the effect of interannual changes on DEM evaluation, the time difference between applied OIB airborne data and ICESat-2 DEM should be less than one year. Thus, OIB airborne data in October and November 2018 and October and November 2019 in Antarctica (Fig. 1a) are chosen to evaluate the accuracy of the ICESat-2 DEM. In order to provide a comprehensive and more robust evaluation of the ICESat-2 DEM, OIB data in areas of low elevation change (i.e., ice sheet interior) from 2009 to 2017 are also used additionally (Fig. 1b). The

CryoSay-2 Low Rate Mode (LRM) mask in Antarctica (which was designed for flat ice sheet interior measurements) is used to extract the regions of low elevation change. CryoSat Geographical Mode Mask (V4.0, updated in 19 August - 26 August 2019) at https://earth.esa.int/eogateway/news/cryosat-geographical-mode-mask-4-0-released is used. The averaged elevation change rate in the used OIB data locations is about -0.0074 ± 0.0821 m/yr from 2003 to 2019, according to elevation change rate estimates from Smith et al. (2020). Hence, we assume that in these areas the effect of the elevation change on the DEM

evaluation can be ignored. Besides, a common OIB data in these areas from 2009 to 2019 (Fig. 1b) are used to provide a robust and reasonable comparison between ICESat-2 DEM and previously published DEMs (see Section 2.3).

In addition, elevation records from kinematic GNSS observations (Schröder et al., 2017) in areas of low elevation change are also used for an additional DEM elevation comparison (Fig. 1c). These GNSS profiles were measured in the region from Vostok Station (106.8°E, 78.5°S) to the East Antarctic coast from 2001 to 2015, an averaged offset of 4.9 cm was found

comparing to OIB airborne data in November 2013. The detailed introduction to the data collection, data processing method and accuracy evaluation can be referred to Schröder et al. (2017).

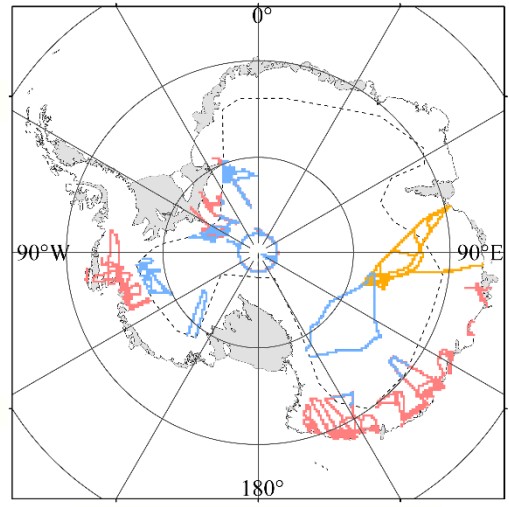



**Figure 1.** Maps of the OIB airborne data in October and November 2018 and October and November 2019 (red), from 2009
to 2019 in ice sheet interior (blue). Map of the GNSS transects from 2001 to 2015 in Antarctica (yellow). The dashed lines
show the boundary of region where we assume to have low elevation change, it is the mode mask boundary of CryoSat-2
LRM data in Antarctica.

## 2.3 Previously published Antarctic DEMs

Six previously published Antarctic DEM products are compared to the ICESat-2 DEM, i.e., ICESat DEM (DiMarzio et al.,
2007), ICESat/ERS-1 DEM (Bamber et al., 2009), Helm CryoSat-2 DEM (Helm et al., 2014), Slater CryoSat-2 DEM (Slater
et al., 2018), REMA DEM (Howat et al., 2019) and TanDEM-X PolarDEM (Wessel et al., 2021), as shown in Section 4.
Detailed information concerning these DEMs is provided in Table 1, and all DEMs have been referenced to the WGS84
ellipsoid.


**Table 1.** Detailed introductions to six previously published Antarctic DEMs, including the source data, time span of the
source data, spatial posting/resolution.

| DEM | Source data | Time span of applied source data | Spatial posting/resolution |
|---|---|---|---|
| ICESat DEM | ICESat | February 2003 to June 2005 | 500 m |
| ICESat/ERS-1 DEM | ICESat, ERS-1 | 1994-1995, 2003-2008 | 1 km |
| Slater CryoSat-2 DEM | CryoSat-2 | July 2010 to July 2016 | 1 km |
| Helm CryoSat-2 DEM | CryoSat-2 | A full 369-day-long cycle starting January 2012 | 1 km |
| REMA DEM | GeoEye-1, WorldView-1/2/3 | 2009-2017, with most collected in 2015 and 2016 | Variable resolutions, 2 and 8 m |
| TanDEM-X PolarDEM | TerraSAR-X, TanDEM-X | April to November 2013, April to October 2014, mid-2014, July 2016 to September 2017 | 90 m |



### 2.4 ICESat-2 DEM generation method

**2.4.1 Surface elevation and uncertainty estimation**

To separate the various contributions (i.e., local surface terrain and elevation change), following Slater et al. (2018), a model fitting method is applied here. The elevation is estimated using a quadratic function based on the local surface terrain and a time term (Eq. 1). This function is fitted in each grid (at the resolutions of 500 m and 1 km, see following subsection) by using an iterative least-squares fit to all the included elevation measurements. By considering the surface elevation

fluctuations and sub-annual changes, this method tends to obtain more accurate elevation estimates (Flament and Remy, 2012; McMillan et al., 2014).

$$E(x,y,t) = \overline{E} + a_0 x + a_1 y + a_2 x^2 + a_3 y^2 + a_4 xy + a_5(t - t_{May\,2019}) \qquad (1)$$

Where $E$ is the surface elevations derived from ICESat-2 measurement points, $x$ and $y$ are the local surface terrain respectively, $t$ is the time term, and $\overline{E}$ is the DEM value in May 2019.

This method suits ICESat-2 orbit cycle, which samples dense ground tracks comparing to previous satellite altimeters, more measurement points are included in the grid cell and the estimated elevations are more robust. It is possible for a quadratic form to model the topography at the resolutions of 500 m and 1 km and smaller elevation residuals can be found than using a simple linear fit (Flament and Remy, 2012). In addition, model fitting method can provide the estimation of elevation change rate ($a_5$), and the estimate agrees well with accurate elevation change estimations from crossover-point

method (Moholdt et al., 2010), which provides an addition reference for the research of ice dynamics and mass balance.

To reduce the effect of any poor fit, a quality control criterion listed in Table 2 is performed, which includes the number of ICESat-2 measurement points used, the time span of the data used, the root-mean-square deviation (RMSD) of the residuals of fitted elevations, the elevation rate of change and its uncertainty. These criteria are constructed for all grid cells, and thus, there are some elevation gaps in the initial DEM. The remaining gaps are filled by using ordinary kriging interpolation

(semi-variogram model: spherical, nugget: 0, sill: 1652285.953, radius: 10 km), which is widely used for generating previous DEMs (Helm et al., 2014; Slater et al., 2018). During the interpolation process, a search radius of 10 km is applied to obtain neighbouring measurement points. Similar estimation models have also been used in previous studies (Moholdt et al., 2010; Flament and Remy, 2012; McMillan et al, 2014; Konrad et al., 2017; Slater et al., 2018), and the evaluation in Section 3.2 also demonstrates its validity.


**Table 2.** Quality control criteria applied to remove the unrealistic elevations due to the poor fitting performances in each grid cell.

| Parameters | Rules |
| --- | --- |
| The number of ICESat-2 measurement points | $\leq 10$ |
| The time span | $\leq 2$ months |





| | |
|---|---|
| RMSD of the residuals of fitted elevations | ≥ 10 m |
| Elevation change rate | ≥ 10 m/yr |
| The uncertainty of elevation change rate | ≥ 10 m/yr |

The performance of this surface fit method is also affected by the spatial distribution and number of ICESat-2
measurement points. After quality control, $4.69 \times 10^9$ ICESat-2 measurement points from November 2018 to November
2019 that cover all of Antarctica are used. An adequate number of ICESat-2 measurement points in one grid cell is required
to generate valid elevation estimates. Fig. 2 shows the distribution of the numbers of ICESat-2 measurement points used in
individual grid cells (at the resolution of 500 m), which indicates a latitude-dependent pattern. Each grid cell contains
approximately 118 ICESat-2 measurement points on average. In the ice sheet interior, the large coverage of ICESat-2
measurement points provides a complete surface height observation. In the low-latitude region, the numbers of ICESat-2
measurement points are relatively small, the proportion of observed grid cells is reduced, and the representativeness is also
reduced.

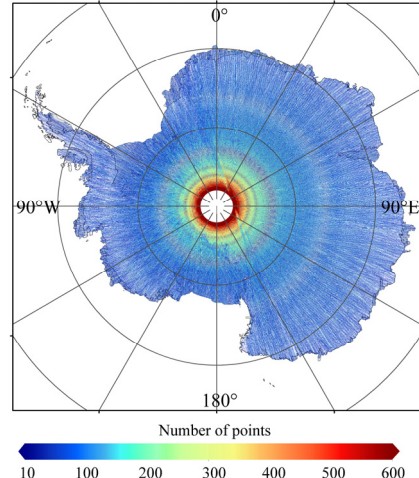

**Figure 2.** Map of the numbers of valid ICESat-2 measurement points in each 500 m grid cell. The numbers of ICESat-2
measurement points in 1 km grid cells are resampled to the resolution of 500 m.

DEM uncertainties are calculated for observed and interpolated grid cells, respectively. The observed grid cell uncertainty
is derived based on the model fitting performance, which is estimated from the equation as follows:

$\mathrm{U} = t(1 - 0.025, n - p) \cdot \mathrm{SE}(\overline{E})$ (2)

where U is the uncertainty of elevation estimate $\overline{E}$, $\mathrm{SE}(\overline{E})$ is the standard error of the elevation estimate $\overline{E}$, and $t(1-0.025,$
$n-p)$ is the 95% percentile of t-distribution with $n$-$p$ degrees of freedom, $n$ is the number of ICESat-2 measurement points in




the grid cell, $p$ is the number of regression coefficients (i.e., 7). For the interpolated grid cells, uncertainty is calculated from the kriging variance error. In the ICESat-2 DEM uncertainty calculation, the uncertainty from ICESat-2 measurements is not considered because the effect of ICESat-2 measurement bias is limited (< 5 cm, Brunt et al., 2019; < 14 cm, Shen et al., 2021).

### 2.4.2 Choice of DEM resolution

The selection criterion of DEM resolution is to present the detailed pattern of elevations and ensure enough spatial coverage of observed elevations (a smaller resolution tends to cause more observed elevation gaps). Although a much finer scale (e.g., 250 m) can reveal a more detailed elevation pattern, this contributes to more gaps among observed elevations. The overall spatial coverages of observed elevations when applying 250 m, 500 m and 1 km resolutions (which are usually applied in the Antarctic DEM) are 26%, 46% and 72%, respectively. High-latitude areas always have higher observed elevation coverages; in lower latitudes there are still some 250 m grid cells with estimated elevations (Fig. 3), however, 250 m DEM only has 26% coverage. The detailed variations in the spatial coverages of observed grid cells at different latitudes at variable spatial resolutions (250 m, 500 m and 1 km, which are usually applied in the Antarctic DEM) are shown in Fig. 4a. 500 m is a reliable grid size which makes denser spatial coverage of the observed elevations, but a single resolution cannot obtain ideal spatial coverage, especially in low-latitude areas. To increase the coverages of observed elevations as much as possible, referring to Slater et al. (2018), two spatial resolutions are used to estimate the surface elevations from ICESat-2. That is, elevations are estimated at resolutions of 500 m and 1 km. The observation gaps in the 500 m DEM are filled by the resampled 1 km DEMs (resampled to the 500 m DEM). The addition of DEMs at 1 km greatly increases the observation coverage, the overall spatial coverage is approximately 74%, and the remaining gaps are filled using ordinary kriging interpolation. Although two resolutions are applied, 1 km and interpolated elevations are both resampled to the posting of 500 m to provide a consistent DEM dataset; hence, the final ICESat-2 DEM is posted at a resolution of 500 m.

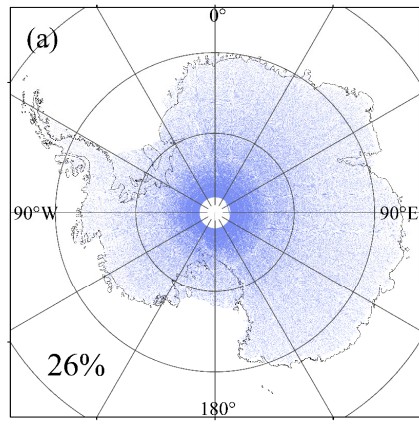
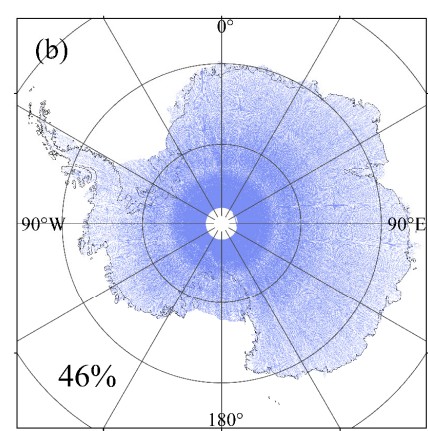
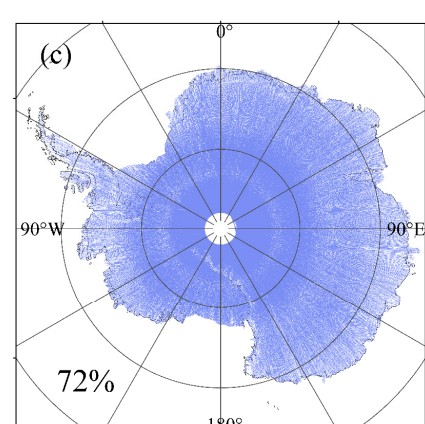

**Figure 3.** Map of the observed grid cells of DEMs at the spatial resolution of 250 m (a), 500 m (b) and 1 km (c). The observed grid cells are coloured in blue, the overall coverage of each DEM in Antarctica is also presented beside.

The application of two resolutions may include additional effects, i.e., different grid cell resolutions tend to present different elevation estimates. Here, we compare the elevation difference at the overlapped areas in Antarctica at different spatial resolutions (Fig. 4b). The elevation values become lower when a larger spatial resolution is applied, which acts as a 'running mean'. Although applying different spatial resolutions affects the elevation values, an averaged elevation difference of 0.04 ± 2.93 m can be found (Fig. 4c), which is quite small comparing to the estimated elevations. In addition, this method can increase the coverage of observed elevations, and observed elevations tend to be more reliable than interpolated elevations (as shown in Section 3.2).

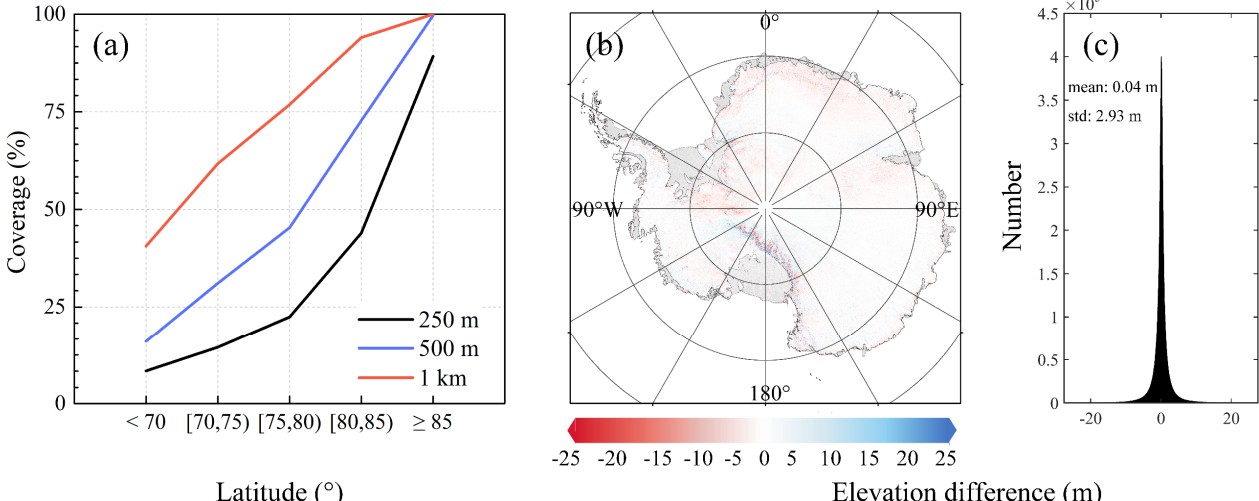

**Figure 4.** (a) Spatial coverages of observed grid cells in the five latitude ranges when three spatial resolutions, i.e., 250 m (yellow), 500 m (blue) and 1 km (red), are applied. (b) Map of the elevation difference of DEMs at the resolutions of 1 km and 500 m. (c) Histograms of the elevation difference of DEMs at the resolutions of 1 km and 500 m, the average and standard deviation values are also presented beside.

### 2.4.3 DEM evaluation method

ICESat-2 DEM and previously published DEMs are resampled to the OIB/GNSS data locations and calculate the difference for evaluation. Four indexes are used to evaluate the DEM performance, including median deviation (MeD), median absolute deviation (MeAD), standard deviation (SD) and RMSD. The corresponding calculation equations are listed as follows:

$$\mathrm{MeD} = \mathrm{median}(\delta_{i=1,2...,n}) \tag{3}$$

$$\mathrm{MeAD} = \mathrm{median}(|\delta_{i=1,2...,n}|) \tag{4}$$



$$SD = \sqrt{\frac{\sum_{i=1}^{n}(\delta_i - MD)^2}{n-1}}$$

(5)

$$RMSD = \sqrt{\frac{\sum_{i=1}^{n}\delta_i^2}{n-1}}$$

(6)

Where $\delta_i$ is the bias of ICESat-2 DEM and OIB/GNSS elevation, MD is the mean deviation and $n$ is the number of the matched grid cells.

### 3 Results

#### 3.1 General attributes of ICESat-2 DEM

The effective time stamp of the ICESat-2 DEM is May 2019, which is halfway between November 2018 and November
2019. The ICESat-2 DEM provides a complete surface elevation reference for Antarctica, which illustrates higher elevations in the ice sheet interior and lower values in marginal ice shelves (Fig. 5). The local slope shows a pattern similar to the DEM, and undulated slopes are found in areas with rugged terrain, such as the Antarctic Peninsula and Transantarctic Mountains (Fig. 6). Both elevation and slope uncertainties show latitude-dependent patterns, and larger values tend to be found at low latitudes, which may be related to the numbers of ICESat-2 measurement points in individual grid cells (Fig. 2).

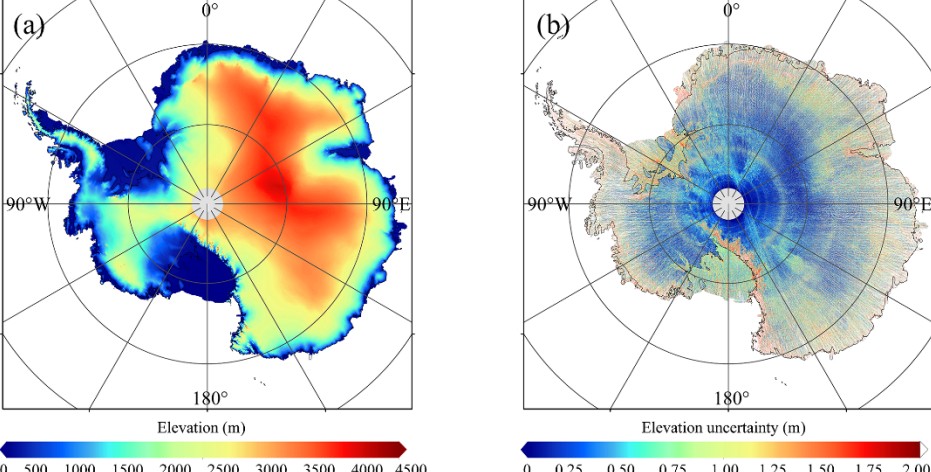

**Figure 5.** (a) A new DEM of Antarctica at a posting of 500 m derived from ICESat-2, which covers both the ice sheet and ice shelves with the southern limit of 88°S. (b) Map of the ICESat-2 DEM elevation uncertainty.

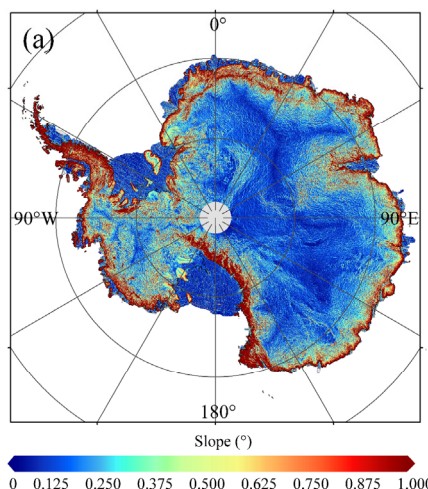

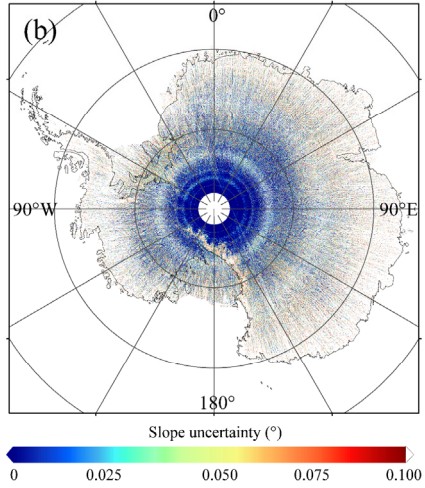

**Figure 6.** (a) Map of the surface slope of Antarctica derived from the ICESat-2 DEM. (b) Map of the ICESat-2 DEM surface slope uncertainty. The uncertainty is estimated based on the propagation of elevation uncertainty.

According to the shaded relief map of Antarctica derived from the ICESat-2 DEM (Fig. 7), obvious topographical patterns and flat terrain can be found in the mountain environments and ice sheet interior, respectively. On the Antarctic Peninsula, the ice shelf limit is visually identified from the shaded relief map (Fig. 7b). Other large-scale terrain features, e.g., subglacial lakes and floating ice shelves, can also be visually detected (Figs. 7c and 7d).

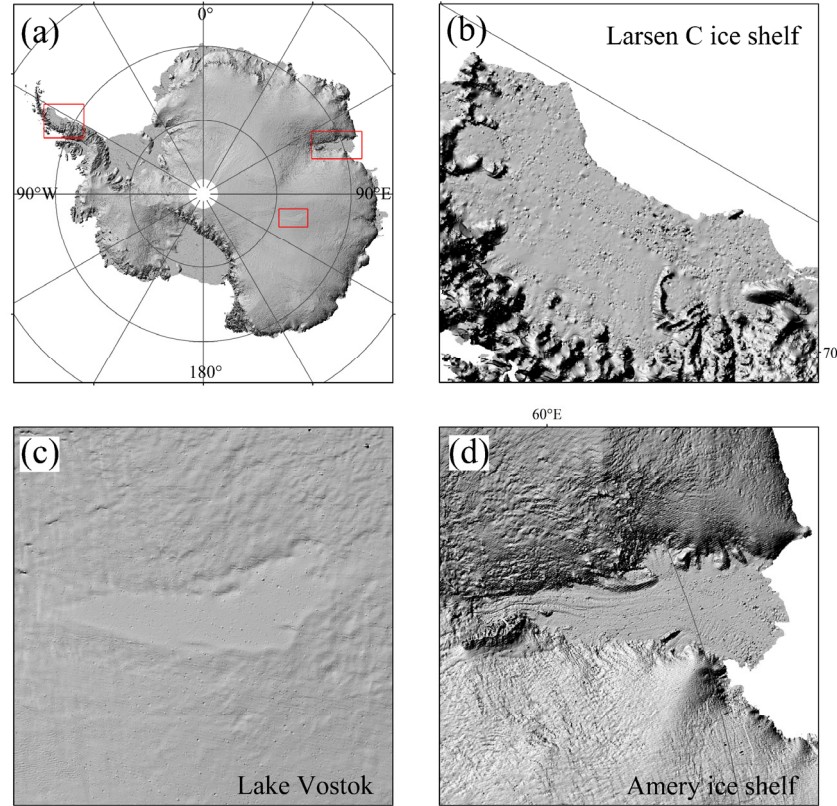

**Figure 7.** (a) Shaded relief map of Antarctica derived from the ICESat-2 DEM. The detailed maps of the Larsen C ice shelf,
Lake Vostok and Amery ice shelf are shown in (b), (c) and (d), respectively, and their locations are also shown in (a) by red
rectangular boxes.

Two spatial resolutions are used in the ICESat-2 DEM, and the distributions of three kinds of grid cells (observed at
individual resolutions and interpolated) show obvious latitude-dependent patterns. Regardless of whether at the basin scale
or regional scale, more elevations at higher resolutions tend to be located in high-altitude areas, while elevations at lower or
interpolated resolutions are mostly located in low-altitude regions (Fig. 8).





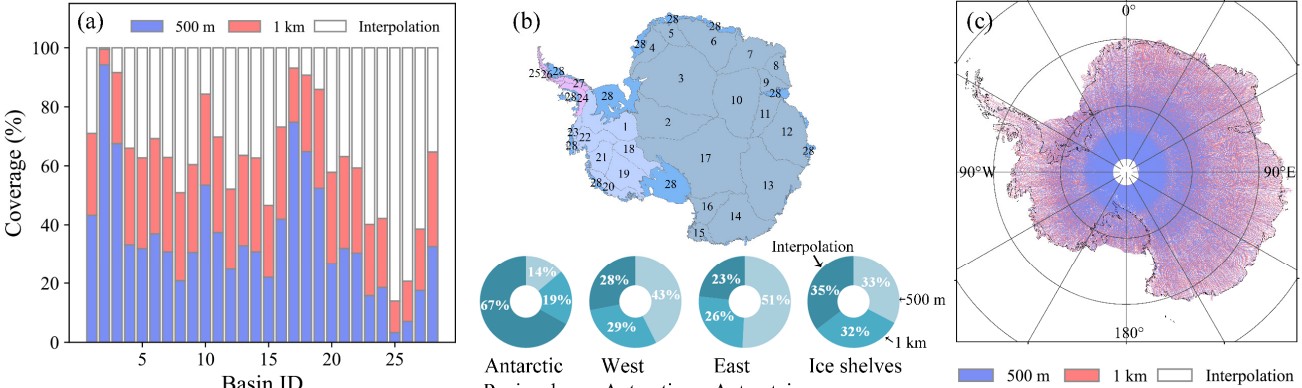

**Figure 8.** (a) Coverages of observed grid cells at 500 m and 1 km and interpolated grid cells in 27 drainage basins of ice sheets (Zwally et al., 2012) and ice shelves. The boundaries and basin index (ID) of 27 ice sheet drainage basins (Numbers 1 to 27) and ice shelves (Number 28) are shown in (b). The coverages of observed (at two spatial resolutions) and interpolated grid cells in the Antarctic Peninsula, West Antarctica, East Antarctica and ice shelves are also shown in (b). (c) Map of the selected grid cell resolution for deriving the ICESat-2 DEM in all grid cells at a spatial resolution of 500 m. Elevation values derived from 1 km and interpolation (i.e., 1 km) are resampled to a resolution of 500 m.

## 3.2 Evaluation of ICESat-2 DEM by comparing to OIB airborne data

In total, approximately $5.2 \times 10^6$ OIB measurement points that cover both the steep and flat regions (Figs. 1a and b) are chosen to evaluate the ICESat-2 DEM. Generally, a MeAD of 1.49 m and an RMSD of 15.64 m are found for ICESat-2 DEM comparing to OIB surface heights (Table 3). Ice sheet elevations are more accurate than those estimated for ice shelves, which may due to a higher percentage of high-slope areas in ice shelves.

**Table 3.** Comparisons between the ICESat-2 DEM and OIB airborne elevation measurements (including data in areas of low elevation change from 2009 to 2017 and data in the Antarctica from 2018 to 2019) in observed and interpolated areas for individual regions (i.e., the ice sheet and ice shelves). MeD: median deviation, MeAD: median absolute deviation, SD: standard deviation, RMSD: root-mean-square deviation.

|  | Region | MeD (m) | MeAD (m) | SD (m) | RMSD (m) | Number of used OIB measurement points |
|---|---|---|---|---|---|---|
| Observed | Ice sheet | 0.08 | 1.18 | 12.75 | 12.75 | 3589087 |
|  | Ice shelves | 0.77 | 2.60 | 15.26 | 15.27 | 191754 |
|  | Total | 0.09 | 1.23 | 12.89 | 12.89 | 3780841 |
| Interpolated | Ice sheet | -0.40 | 2.50 | 20.68 | 20.73 | 1237416 |
|  | Ice shelves | 0.36 | 3.23 | 24.61 | 24.65 | 185613 |
|  | Total | -0.33 | 2.58 | 21.25 | 21.28 | 1423029 |



| Overall | Ice sheet | 0.01 | 1.41 | 15.20 | 15.20 | 4826503 |
|---------|-----------|------|------|-------|-------|---------|
|  | Ice shelves | 0.59 | 2.88 | 20.40 | 20.43 | 377367 |
|  | Total | 0.03 | 1.49 | 15.64 | 15.64 | 5203870 |

We also evaluate the elevation performance for observed and interpolated grid cells (Table 3). Generally, the bias of observed elevations is smaller than that of interpolated elevations in both ice sheets and ice shelves, which indicates that the observed elevations tend to be more accurate than those estimated from interpolation. Larger biases will be included in the ICESat-2 DEM if the coverage of interpolated elevations is high, hence the elevation gaps in the 500 m DEM are firstly

filled by the resampled 1 km DEM to reduce the coverage of interpolated elevations. The accuracy of the ICESat-2 DEM has an obvious relationship with local terrain conditions, and the bias rises when the slope or roughness becomes larger, which is visible for three surface types (Table 4) and different surface slope conditions (Fig. 9). The bias in rocks is obviously larger than those for snow/firn and blue ice areas (BIAs), which is mainly due to the local terrain condition, as they are mostly located in the Transantarctic Mountains and the Antarctic Peninsula, while snow/firn and BIAs tend to have flat surface

terrain; hence, they have smaller biases. While in the low-slope regions, the ICESat-2 DEM shows good agreement with both the OIB and GNSS data; in the large-slope areas, larger biases occur (Fig. 9).

**Table 4.** Comparison between the ICESat-2 DEM and OIB airborne elevation measurements (including data in areas of low elevation change from 2009 to 2017 and data in the Antarctica from 2018 to 2019) with respect to three surface types, i.e.,
snow/firn, blue ice areas (BIAs) and rocks. The surface type data are obtained from Hui et al. (2017).

|  | MeD (m) | MeAD (m) | SD (m) | RMSD (m) | Number of compared grid cells |
|---|---------|----------|--------|----------|-------------------------------|
| Snow/firn | 0.03 | 1.43 | 15.03 | 15.03 | 5046581 |
| BIA | -1.54 | 8.18 | 22.09 | 22.30 | 151111 |
| Rock | -6.88 | 29.62 | 89.65 | 96.84 | 6178 |
| Total | 0.03 | 1.49 | 15.64 | 15.64 | 5203870 |

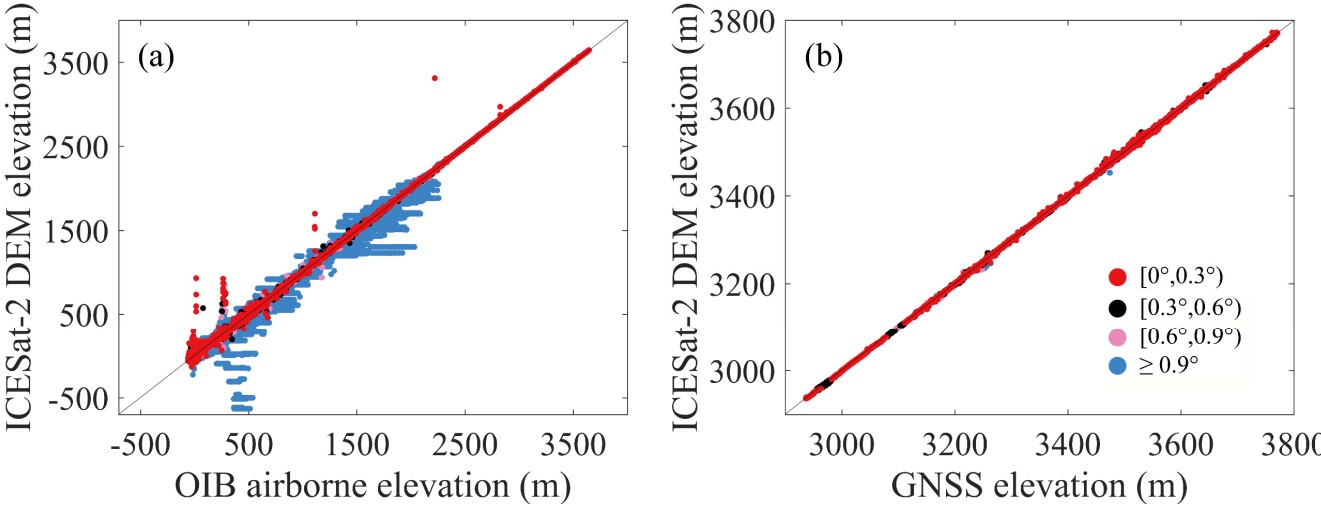

**Figure 9.** Scatter plots of ICESat-2 DEM elevation and OIB airborne elevation (a) and GNSS elevation (b), respectively.
The surface slopes are distinguished in different colours, as shown in the figure legend.

Although OIB airborne data provide an independent evaluation of the generated DEM, they still cannot present a comprehensive comparison. Most of the OIB airborne data were obtained in ice sheet margins or mountain environments, with high slopes and low elevations. Approximately 78% of used OIB elevations are less than 1500 m, and 76% of the observed surface slopes from the OIB mission are less than 1°, while the corresponding percentages from the ICESat-2 DEM are 37% and 89%, respectively. The applied OIB airborne data cannot completely represent the slope/elevation distributions of the Antarctic DEM; hence, the real accuracy of the ICESat-2 DEM is biased and may be higher.

In order to evaluate the DEM performance in more detail, the elevations along two OIB tracks in flat ice sheet interior and rough ice sheet margins are shown in Fig. 10. In the ice sheet interior where surface slopes are small (Fig. 10a), elevation differences of approximately 5 m can be found (the averaged elevation differences for ICESat-2 DEM, ICESat/ERS-1 DEM, Slater CryoSat-2 DEM, Helm CryoSat-2 DEM, REMA DEM and TanDEM-X PolarDEM are 0.03±1.01 m, 49.46±28.53 m, 0.02±4.16 m, -0.06±4.52 m, 0.20±2.17 m and -4.12±1.09 m). The elevation differences are further reduced when surface slope become smaller. While at the Pine Island Glacier where surface slopes are large (Fig. 10b), elevation differences of approximately 20 m can be found in the undulated terrains (the averaged elevation differences for ICESat-2 DEM, ICESat DEM, ICESat/ERS-1 DEM, Slater CryoSat-2 DEM, Helm CryoSat-2 DEM, REMA DEM and TanDEM-X PolarDEM are -0.40±19.43 m, 1.92±27.28 m, 1.24±14.20 m, 0.09±15.34 m, 2.69±13.67 m, 0.32±1.10 m and -0.99±0.92 m). Overall, ICESat-2 DEM has better performances in the flat regions than steep areas. Regions of low surface slope represent the majority of Antarctic ice sheet, hence most elevations from ICESat-2 DEM have smaller elevation biases.


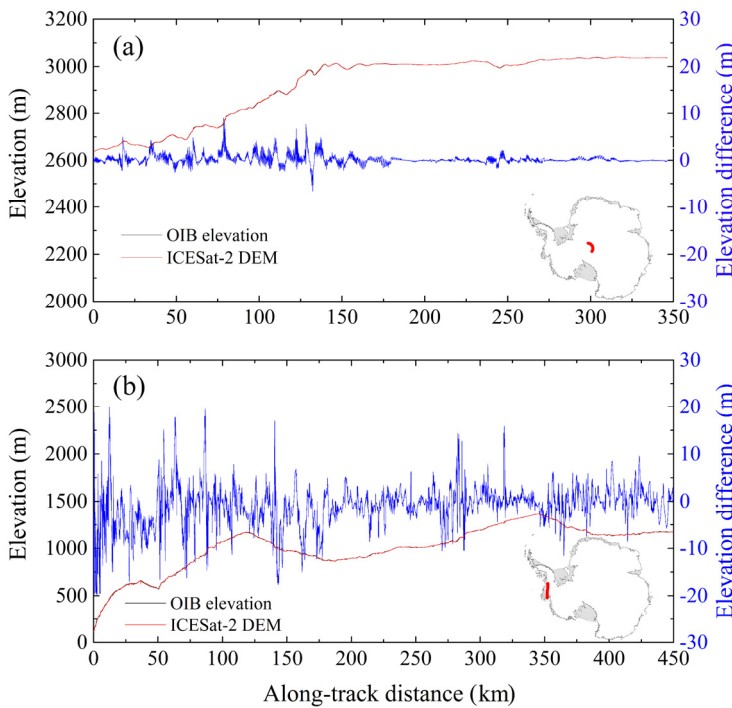

**Figure 10.** Differences between the ICESat-2 DEM and OIB elevations along two OIB flight paths in (a) ice sheet interior and (b) Pine Island Glacier. ICESat-2 DEM elevations are in red, OIB elevations are in black, and the elevation differences between ICESat-2 DEM and OIB elevations are in blue. Locations of the two OIB flight paths are shown in red in the inserted figures of Antarctica.

Additionally, by comparing to the OIB or GNSS elevation data (see Section 4, Table 6), we can estimate the actual ICESat-2 DEM uncertainty as the SD of the differences to OIB or GNSS elevation data. In the estimated uncertainty map (Fig. 5b), 73% grid cells have uncertainty values of < 3m. Regions of lower surface slope which represent the majority of the Antarctic ice sheet – falls typically in the elevation uncertainty range < 3 m. The SD of differences to GNSS data (which were obtained in the low-slope regions) shows a value of 1.67 m (Table 6), indicating that the uncertainty map can represent this. Large uncertainty values (i.e., > 20 m) can be found in the ice sheet margins where some OIB airborne data were obtained, considering that other OIB data were obtained in the regions of low slope the estimated uncertainty map (Fig. 5b) still matches with the actual DEM uncertainty of 15.64 m (estimated from OIB data, Table 3). The estimated uncertainty values can represent the SDs from what is given as OIB and GNSS data, which means that the provided uncertainty estimates are reliable.

## 4 Comparisons with previous published Antarctic DEMs

When compared to the altimeter-derived DEMs, the elevation difference rises when the surface slope becomes larger, especially in mountainous environments (e.g., Transantarctic Mountains and Antarctic Peninsula, Fig. 11). This may be due





to their differences in spatial resolution and measurement accuracy; this effect is considerably reduced when the local terrain
is flatter (e.g., ice sheet interior).

Compared to the REMA DEM and TanDEM PolarDEM, smaller elevation differences can be found in both the flat ice
sheet interior and steep mountains/marginal ice sheets. As shown in Table 5, REMA DEM and TanDEM PolarDEM are
more accurate than altimeter-derived DEMs; hence, similar elevations indicate the reliability of ICESat-2 DEM in mountain
environments. In particular, the ICESat-2 DEM shows a generally higher surface height than the TanDEM PolarDEM, which
is assumed to be caused by the penetration depth of the X-band (TerraSAR-X and TanDEM-X) into snowpack (Dehecq et al.,
2016; Fischer et al., 2020).

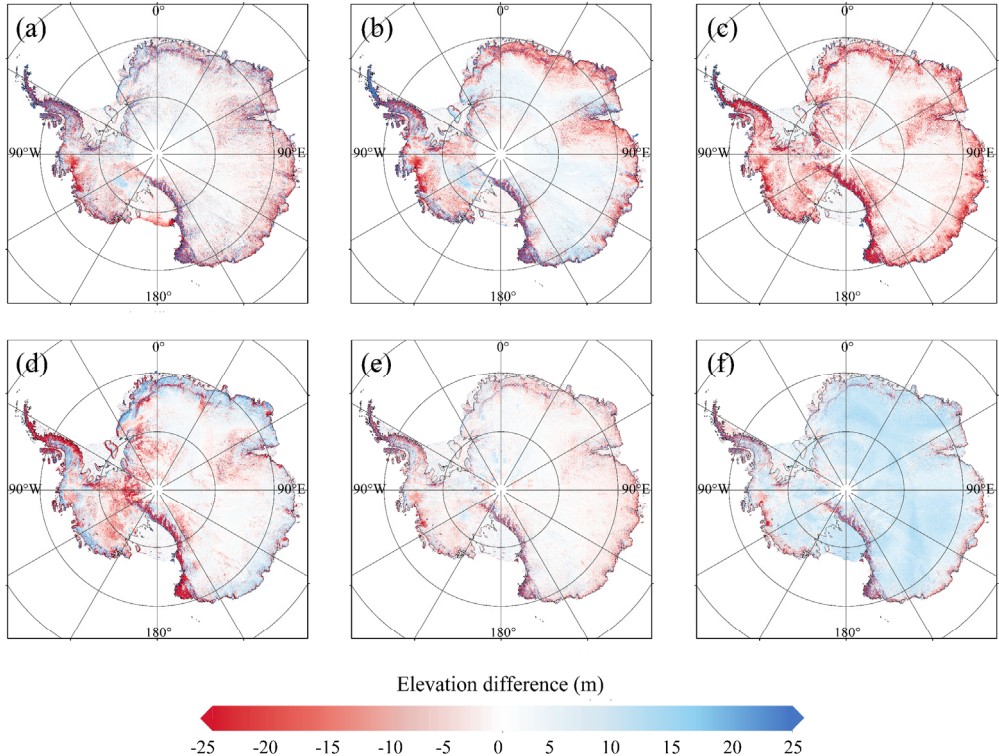

**Figure 11.** Elevation differences between the ICESat-2 DEM and six previously published DEMs, i.e., ICESat DEM (a),
ICESat/ERS-1 DEM (b), Helm CryoSat-2 DEM (c), Slater CryoSat-2 DEM (d), REMA DEM (e) and TanDEM PolarDEM
(f).

To indicate a fair and quantitative comparison between the ICESat-2 DEM and other DEMs, OIB airborne data in areas of
low elevation change from 2009 to 2019 are used to evaluate individual DEMs, and the same evaluation method applied for
the ICESat-2 DEM is used (as described in Section 2.2). The evaluation result shows that the ICESat-2 DEM has a better
performance than altimeter-derived DEMs and is comparable to the DEMs derived from stereo-photogrammetry and
interferometry (Table 5).

**Table 5.** Comparisons between the ICESat-2 DEM, ICESat DEM, ICESat/ERS-1 DEM, Helm CryoSat-2 DEM, Slater CryoSat-2 DEM, REMA DEM, TanDEM PolarDEM and OIB airborne elevation measurements in areas of low elevation change from 2009 to 2019.

|  | MeD (m) | MeAD (m) | SD (m) | RMSD (m) | Number of used OIB measurement points |
|---|---|---|---|---|---|
| ICESat-2 DEM | 0.10 | 0.98 | 5.36 | 5.38 | |
| ICESat DEM | -2.61 | 6.35 | 19.90 | 20.43 | |
| ICESat/ERS-1 DEM | -0.15 | 1.84 | 11.53 | 11.54 | |
| Helm CryoSat-2 DEM | 0.65 | 2.68 | 24.97 | 25.02 | 1965309 |
| Slater CryoSat-2 DEM | 1.22 | 2.87 | 23.85 | 24.14 | |
| REMA DEM | -0.16 | 0.53 | 1.75 | 1.76 | |
| TanDEM PolarDEM | -2.84 | 2.94 | 2.76 | 3.90 | |

The median differences in surface slope and roughness for these five DEMs illustrate that all their elevation biases become more uncertain with increasing slope and roughness (Fig. 12). The ICESat-2 DEM outperforms other altimeter-derived DEMs for all surface conditions. The REMA DEM always has more stable performances than the ICESat-2 DEM, as stereo-photogrammetry can generate more consistent elevation estimations at the regional scale than altimetry. A similar situation occurs for the TanDEM PolarDEM when slopes >1.5°. Nevertheless, the ICESat-2 DEM is comparable to both the REMA DEM and TanDEM PolarDEM when slopes are less than 1°, which occupies 89% of Antarctica north of 88°S.

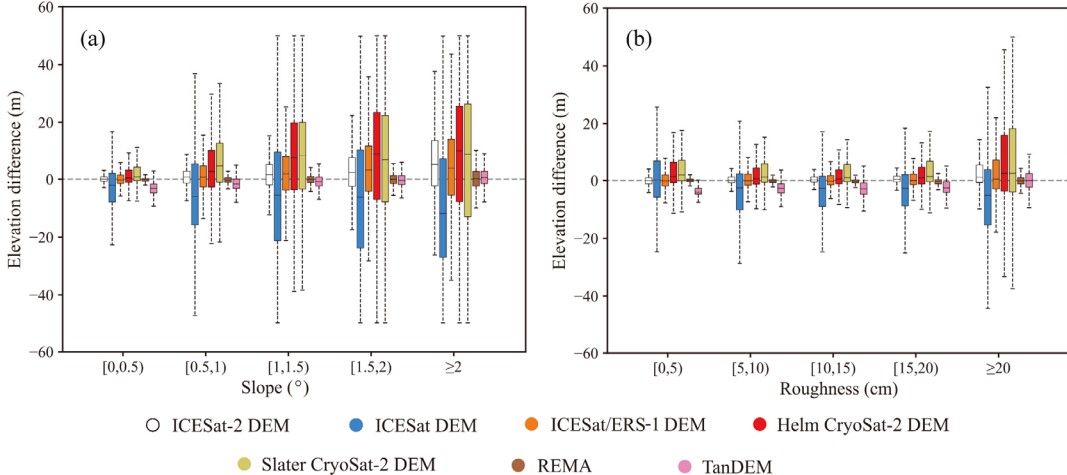

**Figure 12.** Median differences between seven DEMs and OIB airborne elevation measurements in areas of low elevation change from 2009 to 2019 with respect to surface slope and roughness. The upper and lower lines in each box indicate the





25th and 75th percentiles, the whiskers indicate the 5th and 95th percentiles, and the central horizontal line indicates the median difference.

Here, kinematic GNSS data from 2001 to 2015 in ice sheet interior are used to construct an additional DEM evaluation. As surfaces in the interior of East Antarctica are flat, better performances for all DEM except TanDEM PolarDEM are found than these based on OIB airborne data. Similarly, ICESat-2 DEM outperforms all altimeter-derived DEMs and TanDEM
PolarDEM, and is comparable to REMA DEM (Table 6). Additionally, the accuracy of the ICESat-2 DEM is related to the surface slope (Fig. 9b). However, as the terrain conditions of GNSS measurement points are relatively flat, this relationship is not obvious.

**Table 6.** Comparisons between the ICESat-2 DEM, ICESat DEM, ICESat/ERS-1 DEM, Helm CryoSat-2 DEM, Slater
CryoSat-2 DEM, REMA DEM, TanDEM PolarDEM and GNSS elevation data in areas of low elevation change from 2001 to 2015.

|  | MeD (m) | MeAD (m) | SD (m) | RMSD (m) | Number of used OIB measurement points |
|---|---|---|---|---|---|
| ICESat-2 DEM | -0.03 | 0.41 | 1.17 | 1.17 | |
| ICESat DEM | -1.91 | 2.89 | 5.21 | 5.97 | |
| ICESat/ERS-1 DEM | -0.74 | 0.84 | 1.39 | 1.61 | |
| Helm CryoSat-2 DEM | 0.07 | 0.67 | 1.67 | 1.71 | 488963 |
| Slater CryoSat-2 DEM | 0.00 | 0.46 | 1.65 | 1.66 | |
| REMA DEM | 0.03 | 0.26 | 0.57 | 0.57 | |
| TanDEM PolarDEM | -4.62 | 4.62 | 1.33 | 4.72 | |

Although the derived ICESat-2 DEM is less accurate than REMA DEM and TanDEM PolarDEM, considering the measurement accuracy of altimetry, these differences are still acceptable. Elevation change rate can be obtained when
deriving the ICESat-2 DEM, which can provide an additional reference for ice topography and mass balance estimation. Comparing to altimeter-derived DEMs, ICESat-2 DEM has better (or comparable) performance in accuracy, resolution and coverage.

In previous studies, several years of altimeter data are needed to derive the DEM in Antarctica. Due to the high-density measurements of ICESat-2, 13 months of ICESat-2 data can be used to generate a DEM for Antarctica, and the elevation
accuracy is superior than other altimeter-derived DEMs. This means that the ICESat-2 DEM can be updated annually. This study demonstrates the feasibility and reliability of using one-year ICESat-2 data to derive the Antarctic DEM, provides a reference for the processing scheme of DEM (e.g., in higher resolution, regularly updated) based on ICESat-2 in future.

**5 Data availability**

The generated ICESat-2 DEM (including the map of uncertainty) can be downloaded from National Tibetan Plateau Data
Center, Institute of Tibetan Plateau Research, Chinese Academy of Sciences at https://data.tpdc.ac.cn/en/disallow/9427069c-
117e-4ff8-96e0-4b18eb7782cb/ (Shen et al., 2021, DOI: 10.11888/Geogra.tpdc.271448).

**6 Conclusions**

A new DEM for Antarctica with a posting of 500 m is presented based on the surface height measurements from ICESat-2
by using a model fitting method. This DEM has an elevation measurement that accounts for 74% of Antarctica, and the
remaining 26% is estimated based on the ordinary kriging method. The accuracy of the ICESat-2 DEM is evaluated by
comparing it to the independent airborne data from the OIB mission. Overall, the ICESat-2 DEM shows a median bias of
0.03 m and an RMSD of 15.64 m, and these accuracies are compromises for DEM values from surface fits and interpolation.
A median bias of 0.09 m and an RMSD of 12.89 m are found for areas where elevations are derived from ICESat-2
measurements, and they increase to -0.33 m and 21.28 m for interpolated elevations. The accuracy decreases when the
surface slope or roughness increases; thus, larger biases occur for steep rocks, and flat snow/firn and blue ice areas have
smaller elevation differences.

Compared to DEMs derived from satellite altimeters (i.e., the ICESat DEM, ICESat/ERS-1 DEM, Helm CryoSat-2 DEM,
and Slater CryoSat-2 DEM), larger differences are found in regions with high slopes, which is due to their resolution
difference, while smaller elevation differences compared to the REMA DEM and TanDEM PolarDEM support the reliability
of the ICESat-2 DEM. Based on the OIB airborne data and kinematic GNSS transects, the ICESat-2 DEM shows better
performance than altimeter-derived DEMs and is comparable to the fine-scale REMA DEM and TanDEM PolarDEM, which
demonstrates the reliability of the ICESat-2 DEM. More importantly, this study demonstrates that the ICESat-2 DEM can be
updated annually, and elevation change rate can also be obtained when deriving the ICESat-2 DEM, which can provide an
additional reference for ice topography and mass balance estimation.

**Author contribution**

Xiaoyi Shen and Yubin Fan developed the related algorithm, generated and evaluated the ICESat-2 DEM; Lhakpa Drolma
constructed the comparison to previously published DEM products; Chang-Qing Ke supervised this work.

**Competing interests**

The authors declare that they have no conflict of interest



## Acknowledgments

This work is supported by the Programs for National Natural Science Foundation of China [grant numbers 41976212, 41830105].

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
