# Peer review of "A new digital elevation model (DEM) dataset of the entire Antarctica continent derived from ICESat-2"

_Earth System Science Data, 2021_

## Author Comment (AC1)

We thank the reviewer for the helpful feedback, these suggestions have significantly improved the text and figures, we are appreciative of the help and time.

We have addressed all the comments here, point by point responses to the comments are listed in BLUE.

Here we summarize the major revision in the revised manuscript:

According to the reviewers, additional filters (a 3-standard-deviation filter and a median filter) have been applied in the generated DEM to improve the DEM performance. The new DEM was also evaluated by comparing to the OIB and GNSS data, similar performances can be found (as listed in the Tables below). In particular, to account for the temporal difference between the DEM and OIB/GNSS data, when performing the comparisons we adjusted the ICESat-2 DEM elevations for the surface elevation changes between the acquisition periods of these two data. The adjustments were calculated by using the trend values derived from Smith et al. (2020) and we assumed the constant elevation change rates, these were applied for the DEM values in the locations of OIB/GNSS measurements. The related text, figures and tables have been accordingly revised, the same conclusions are derived.

The updated DEM map (including uncertainty map) can be downloaded from Google drive at https://drive.google.com/drive/folders/1h0-QxAkjGMSc-eqlBBigiqvgp0k-8iIB?usp=sharing at this stage, and we will replace the previous revision in the data storage (i.e., National Tibetan Plateau Data Center, Institute of Tibetan Plateau Research, Chinese Academy of Sciences) after the manuscript revision.

Table 4 (previous Table 3). Comparisons between the ICESat-2 DEM and OIB airborne elevation measurements (including data in areas of low elevation change from 2009 to 2017 and data in the Antarctica from 2018 to 2019) in observed and interpolated areas for individual regions (i.e., the ice sheet and ice shelves). MeD: median deviation, MeAD: median absolute deviation, SD: standard deviation, RMSD: root-mean-square deviation.

| | Region | MeD (m) | MeAD (m) | SD (m) | RMSD (m) | Number of used OIB measurement points |
|---|---|---|---|---|---|---|
| Observed | Ice sheet | -0.17 | 1.21 | 9.25 | 9.26 | 3589087 |
| | Ice shelves | 0.59 | 2.53 | 14.07 | 14.09 | 191754 |
| | Total | -0.15 | 1.26 | 9.56 | 9.57 | 3780841 |
| Interpolated | Ice sheet | -0.52 | 2.63 | 13.30 | 13.36 | 1237416 |
| | Ice shelves | 0.44 | 3.00 | 15.16 | 15.21 | 185613 |
| | Total | -0.41 | 2.67 | 13.58 | 13.62 | 1423029 |
| Overall | Ice sheet | -0.22 | 1.47 | 10.44 | 10.47 | 4826503 |
| | Ice shelves | 0.53 | 2.75 | 14.62 | 14.65 | 377367 |
| | Total | -0.19 | 1.54 | 10.81 | 10.83 | 5203870 |

Table 7 (previous Table 6). Comparisons between the ICESat-2 DEM, ICESat DEM, ICESat/ERS-1 DEM, Helm CryoSat-2 DEM, Slater CryoSat-2 DEM, REMA DEM, TanDEM PolarDEM and

GNSS elevation data in areas of low elevation change from 2001 to 2015.

| | MeD (m) | MeAD (m) | SD (m) | RMSD (m) | Number of used GNSS measurement points |
|---|---|---|---|---|---|
| ICESat-2 DEM | 0.02 | 0.50 | 1.59 | 1.60 | |
| ICESat DEM | -3.79 | 4.30 | 10.99 | 13.10 | |
| ICESat/ERS-1 DEM | -0.75 | 1.02 | 2.22 | 2.32 | |
| Helm CryoSat-2 DEM | 0.16 | 0.89 | 2.84 | 2.92 | 488963 |
| Slater CryoSat-2 DEM | -0.12 | 0.61 | 2.41 | 2.43 | |
| REMA DEM | 0.06 | 0.30 | 0.78 | 0.78 | |
| TanDEM PolarDEM | -4.03 | 4.03 | 1.52 | 4.34 | |

First of all, I would suggest a different paper title to reflect the scope of the ESSD, e.g. "A new digital elevation model (DEM) dataset of the entire Antarctica continent derived from ICESat-2"

Agree and accept.

This manuscript provides an Antarctic DEM data set based on NASA's new generation of ICESat-2 altimeter. The authors applied the spatiotemporal fitting method, so the data set covers both the ice sheet and ice shelves. This is not the first manuscript to tackle the DEM data set for the Antarctic. Nevertheless, the authors have demonstrated their product and evaluated it using the OIB and GNSS data under various surface conditions. As far as I understood, the ICESat-2-derived Antarctic DEM is not available. Considering the high-resolution and accurate measurements of ICESat-2, I expect this dataset could be valuable for Antarctic glacier research. For this reason, I would like to see this paper to be published.

In general, this paper is well written, and the structure is clear and easy to follow. It is an interesting topic and is worth to be published as ICESat-2 provides elevation measurements in much higher spatial-temporal resolutions. One weak point I can tell is that the authors provide just a 1-year data set. On the other hand, think of the entire Antarctic domain, one year data set is already quite comprehensive, in particular, authors have claimed that they can provide annual data sets in a sustainable way meaning the data set can be accumulated on an annual base.

Thank you very much for your positive feedback and advice.

I think the conclusion is rather short, I would like to see recommendations that authors could point out, e.g., a number of potential applications applying this data set or the forthcoming new DEM data sets generated by the methodology authors have applied. This improvement would strengthen this data paper.

Accept. One paragraph has been added in the Section Conclusions to point out the potential applications of our data set:'… *Here thirteen-months of ICESat-2 data are used to generate the Antarctic DEM and the evaluation result shows that the corresponding DEM is reasonable and valid. This means that the ICESat-2 DEM can be provided in a sustainable way, i.e., this DEM can be*

*updated annually and thus accumulated on an annual base. Additionally, reasonable elevation-change rates can also be obtained when deriving the DEM. The combination of the derived DEMs and elevation-change rates can be further used for the references of fieldwork planning, ice motion tracking, numerical modelling of ice sheet and the mass balance estimation. More importantly, this data can be provided on an annual based, which has large application potential for Antarctic research especially under the warm climate.'.*

Some specific comments I hope authors may find useful:

Abstract: "Antarctic digital elevation models (DEMs) data sets are essential,,,"; "human fieldwork", is there any nonhuman fieldwork?

This has been changed to '… are essential for  fieldwork'.

Introduction: P3, L80: Do you apply any other quality control criteria than what you have mentioned here?

No, we only used the data quality flag in ATL06 data (i.e., the surface signal confidence metric) to filter the data with bad quality. Besides, a series of quality control criteria were applied for the DEM estimation, as shown in Table 2.

P3, L84: "Although the signal energies of strong and weak beams are different, all six beams provide centimetre-scale elevation measurements, and the biases of two beams in one pair are less than 2 cm (Brunt et al., 2019) and 5 cm (Shen et al., 2021) for flat and steep surfaces. Thus, the effect of elevations estimated from weak beams is negligible" Not very clear text, please explain more in detail.

We have revised this sentence to make a clear expression: '*Although the signal energies of strong and weak beams are different, all six beams provide centimetre-scale elevation measurements, and the biases of two beams in one pair **are less than 2 cm (Brunt et al., 2019) for flat regions and 5 cm (Shen et al., 2021) for steep surfaces**. Thus, the effect of elevations estimated from weak beams is negligible*'.

P4, L92: "Icessn" ?

IceBridge ATM L2 Icessn elevation, slope and roughness (V002) product (Studinger et al., 2014) is used here for DEM evaluation. According to Studinger et al. (2014), '*… the fundamental form of ATM topography data is a sequence of laser footprint locations acquired in a swath along the aircraft flight track. The **icessn program** condenses the ATM surface elevation measurements by fitting a plane to blocks of points selected at regular intervals along track and several across track. …*'.
Here, Icessn is a terminology.

Reference:

Studinger, M.: IceBridge ATM L2 Icessn Elevation, Slope, and Roughness, version 2. Boulder, Colorado USA: National Snow and Ice Data Center, Digital media, https://doi.org/10.5067/CPRXXK3F39RV, 2014.

P4, L95: What do you mean by 'the effect of interannual changes' here?

It means *the effect of interannual changes of surface elevations*, and we have revised this in the revised manuscript.

P6, L137: this model-fitting method has been used in other papers (e.g., Slater et al., 2018). They have produced multi-annual data, while in this paper you have made just one year of data. Can you point out the differentiation between your work and theirs, e.g., does the length of data processing matter?

The method in Slater et al. (2018) was used here to generate the DEM from ICESat-2 data, both the elevation measurement data were used (CryoSat-2 in Slater et al. (2018) and ICESat-2 in this study). Additionally, as the CryoSat-2 radar signals may penetrate the snow layer, the elevation measurements from ICESat-2 tend to have less uncertainty than those from CryoSat-2 and hence ICESat-2 is expected to have better performance. However, we also notice the difference in the length of data in these two studies. We agree that the spatio-temporal fitting method may be more appropriate for longer time series of altimeter data. However, if this method can still separate temporal elevation changes with just one year of data, it still can be used for DEM generation. Hereafter we provide the map of elevation change rate estimated from this study, we also provide the estimation result from 2003 to 2019 in Smith et al. (2020) for a comparison. Overall, considering the time difference similar elevation change patterns can be found between the two figures. For example, larger elevation decreases can be found in the margin of West Antarctica, obvious elevation increases can be found in the interior of West Antarctica (red cycles in the figure). The elevation change pattern based on one-year ICESat-2 data is reasonable, which indicates that one year of data can give a reliable elevation change map and the elevation estimation is thus reliable. This may due to the much higher measurements density and accuracy of ICESat-2 than previous altimeters. In addition, ICESat-2 DEM has a comparable performance to other DEMs by comparing to the same airborne and GNSS data sets, which also proves the feasibility of the data and method.

[Figure]

**Figure.** Map of elevation change rate in Antarctica derived from one year of ICESat-2 data in this study (left) and map of elevation change rate in Antarctica from 2003 to 2019 in Smith et al. (2020) (right).

One-year of satellite altimeter data have also been used to generate the DEM for Antarctica in previous studies. In Helm et al. (2014), one-year of CryoSat-2 data were used for the DEM generation. The DEM is in the spatial resolution of 1 km and the gaps were filled by using the ordinary kriging interpolation (also a series of processing scheme was included), a quite good performance can be found when comparing to the ICESat elevation data. ICESat-2 has much denser and larger coverage than CryoSat-2, hence it is also reasonable to derive the Antarctic DEM by using one-year of ICESat-2 data considering the performance of one-year CryoSat-2 data. In addition, approximately $4.69 \times 10^9$ ICESat-2 measurement points from November 2018 to November 2019 were used in this study, while $2.5 \times 10^8$ CryoSat-2 measurement points from July 2010 to July 2016 were used in Salter et al. (2018). Considering this, although only one year of ICESat-2 data were used the data density is still larger than seven years of CryoSat-2 data.

Considering the data density/coverage, method performance and DEM accuracy, the DEM generated by using spatio-temporal fitting method from one-year of ICESat-2 data is still reasonable and reliable.

The related statements above have also been added into the revised manuscript (i.e., subsection 2.4.1):
'… *Additionally, the performance of the surface fit method also depends on the timespan of the input data, that is to say, it should be noted that whether one-year of ICESat-2 data can be used to obtain a satisfied fitting performance. Here we find that the elevation-change rate map based on one-year ICESat-2 data (i.e., $a_5$ in Eq.1) has a similar pattern with that from Smith et al. (2020), which estimated the elevation-change rate from 2003 to 2019 based on ICESat and ICESat-2 data, indicating that one-year of data can also provide the reasonable elevation change rates and thus the surface fit method used here is reliable.*'

In addition, in Section 4 we also add a paragraph to point out the differentiation between our work and other studies:

'... *Comparing to other DEMs, elevation change rate can be obtained when deriving the ICESat-2 DEM, which provides an additional reference for ice topography and mass balance estimation. Additionally, in previous studies several years of altimeter data are needed to derive the DEM in Antarctica. Due to the high-density measurements of ICESat-2, 13 months of ICESat-2 data can be used to generate a DEM for Antarctica and the performance is comparable to other DEMs, indicating that the ICESat-2 DEM can be updated annually. This study demonstrates the feasibility and reliability of using one-year ICESat-2 data to derive the Antarctic DEM, provides a reference for the processing scheme of DEM (e.g., in higher resolution, regularly updated) based on ICESat-2 in future.*'.

References:

Smith B, Fricker H A, Gardner A S, et al. Pervasive ice sheet mass loss reflects competing ocean and atmosphere processes. Science, 2020, 368(6496): 1239-1242.

Slater T, Shepherd A, McMillan M, et al. A new digital elevation model of Antarctica derived from CryoSat-2 altimetry. The Cryosphere, 2018, 12(4): 1551-1562.

Helm, V., Humbert, A., and Miller, H.: Elevation and elevation change of Greenland and Antarctica derived from CryoSat-2, The Cryosphere, 8, 1539-1559, https://doi.org/ 10.5194/tc-8-1539-2014, 2014.

Additionally, considering the higher coverage and spatial resolution of ICESat-2, applying a fitting model to ICESat-2 will resolve its finer observations which are not obtained by other satellite altimeters. Can you try to make use of all ICESat-2 data and apply the kriging interpolation directly, in this way you may obtain a more detailed and accurate elevation map, due to the higher resolution and accurate measurements of ICESat-2? The Authors should clearly state why this estimation method is suitable for ICESat-2 data.

A model fitting method used here is to separate the various contributions to the estimated elevations within each grid cell (Flament and Remy, 2012; McMillan et al., 2014), including local surface terrain and elevation change. This function is fitted in each grid cell by using an iterative least-squares fit to all the elevation measurements to minimize the impact of outliers. A quality control criterion is also used to reduce the effect of any poor fit. This method suits ICESat-2 orbit cycle, which samples dense ground tracks comparing to previous satellite radar altimeters, more measurement points are included in the grid cell and the estimated elevations are more robust. The resolutions of grid cells (i.e., 500 m and 1 km) are appropriate for the used ICESat-2 data in this study. Firstly, most elevations (72%) can be directly estimated based on this method. Secondly, it is possible for a quadratic form to model the topography at these scales and smaller elevation residuals can be found than using a simple linear fit (Flament and Remy, 2012).

Approximately $4.69 \times 10^9$ ICESat-2 measurement points are used for elevation estimation in this study, which has a coverage of 18% for the Antarctica. The direct application of kriging interpolation based on all valid measurements means the 72% elevations are estimated from interpolation. As the evaluation results shown in this study and also Slater et al. (2018), the bias of observed elevations

is obviously smaller than that of interpolated elevations, hence the interpolation ratio should be reduced as possible. The model fitting method considers the various contributions to the estimated elevations by including all data acquired within each region, the interpolation ratio is reduced and the derived elevations can represent the elevation in each region well. In addition, model fitting method can provide the estimation of elevation change rate, and the estimate agrees well with accurate elevation change estimations from crossover-point method (Moholdt et al., 2010), which provides an addition reference for the research of ice dynamics and mass balance.

The map for the elevation-change rate ($a_5$) can also prove the reliability of the method, as shown in the comment above.

The above discussions have been listed in the manuscript (see subsections 2.4.1 and 2.4.2).

References:

Moholdt G, Nuth C, Hagen J O, et al. Recent elevation changes of Svalbard glaciers derived from ICESat laser altimetry. Remote Sensing of Environment, 2010, 114(11): 2756-2767.

Flament T, Rémy F. Dynamic thinning of Antarctic glaciers from along-track repeat radar altimetry. Journal of Glaciology, 2012, 58(211): 830-840.

McMillan M, Shepherd A, Sundal A, et al. Increased ice losses from Antarctica detected by CryoSat-2. Geophysical Research Letters, 2014, 41(11): 3899-3905.

Slater T, Shepherd A, McMillan M, et al. A new digital elevation model of Antarctica derived from CryoSat-2 altimetry. The Cryosphere, 2018, 12(4): 1551-1562.

P6, L144: I would like to see a figure for elevation change rate (a5), which can be used to evaluate the method performance. In addition, I have some concerns if one year of data is enough to estimate a reliable elevation change. Could you please provide the elevation change rate map (a5) to see if the method makes sense?

The figure for elevation change rate ($a_5$) and reasons why one-year of ICESat-2 data can still obtain reliable result have been listed in the comments above.

The related discussion has also been added in the Section 2.4.1:
'*… Additionally, the performance of the model fitting method also depends on the amount of the input data, that is to say, it should be noted that whether one-year of ICESat-2 data can be used to obtain a satisfied fitting performance. Here we find that the elevation-change rate map based on one-year ICESat-2 data (i.e., $a_5$ in Eq.1) has a similar pattern with that from Smith et al. (2020), which estimated the elevation-change rate from 2003 to 2019 based on ICESat and ICESat-2 data, indicating that one-year of data can also provide the reasonable elevation change rates and thus the surface fit method used here is reliable.*'.

P8, L185- 195: please explain clearly among those resolution numbers, What exactly number you have finally applied and why?

We have explained it in the subsection 2.4.2: '*The detailed variations in the spatial coverages of*

*observed grid cells at different latitudes at variable spatial resolutions (250 m, 500 m and 1 km, which are usually applied in the Antarctic DEM) are shown in Fig. 4a. 500 m is a reliable grid size which makes denser spatial coverage of the observed elevations, but a single resolution cannot obtain ideal spatial coverage, especially in low-latitude areas. To increase the coverages of observed elevations as much as possible, referring to Slater et al. (2018), two spatial resolutions are used to estimate the surface elevations from ICESat-2.* **That is, elevations are estimated at resolutions of 500 m and 1 km. The observation gaps in the 500 m DEM are filled by the resampled 1 km DEMs (resampled to the 500 m DEM). The addition of DEMs at 1 km greatly increases the observation coverage, the overall spatial coverage is approximately 74%, and the remaining gaps are filled using ordinary kriging interpolation.**'.

P9, L218: why do you use this method, why don't you resample the OIB to the DEM data and calculate the difference and its statistics?

The method (calculating a median or mean OIB elevation for each DEM grid cell) will certainly influence the evaluation results as the DEMs (including previously published DEMs) have different spatial resolutions. Additionally, OIB is the reference elevation and cannot be replaced by the median or mean values, because by calculating a median for each grid cell we assumed that the surface in the grid cell is flat, while in the Eq. 1 we assumed a quadratic surface.

The reason for the choice of this method has been added in the revised text (see subsection 2.4.3): '*… ICESat-2 DEM and previously published DEMs are resampled to the OIB/GNSS data locations and calculate the difference for evaluation,* **to reduce the effect of resolution differences between various DEMs.**'.

Table 3: your uncertainty map shows values of < 2m, in this Table an SD of 15 m can be found, which means that the uncertainty map may not represent this, can you explain this? Additionally, the predicted uncertainty (i.e., uncertainty map) with the actual uncertainty (comparison with OIB and GNSS data) should be compared and discussed.

According to the Reviewer, we estimated the ICESat-2 DEM uncertainty based on another method, which is introduced in the below. The related comparisons and discussion about the predicted uncertainty and actual uncertainty has been revised in the Section 3.2:
'*Additionally, by comparing to the OIB or GNSS elevation data, we can estimate the actual ICESat-2 DEM uncertainty as the SD of the differences to OIB or GNSS elevation data. In the estimated uncertainty map (Fig. 5b), a median value of 5.84 ± 5.29 m can be found. The SD of differences to OIB data which obtained in the large scale shows a value of 10.44 m (Table 4, including plenty of measurements in ice sheet margin), while in the ice sheet interior a value of 3.26 m is found (Table 6). Considering the data coverage and surface-slope difference, the estimated uncertainty values can represent the SDs from what is given as OIB, which means that the provided uncertainty estimates are reliable. Small SD value of 1.59 m can be found when comparing to the GNSS data (Table 7) which were obtained in the regions of low slope, this may due to the resolution and measurement accuracy differences between airborne and GNSS data, hence the ICESat-2 DEM uncertainty map may be slightly overestimated and can be assumed as the upper limit.*'.

DEM uncertainties are calculated based on the approach in Helm et al. (2014). The OIB elevation data are used as the reference and the elevation differences due to the time difference between OIB data and DEM are corrected based on the elevation-change rate from Smith et al. (2020). The DEM uncertainty is then calculated from surface slope, roughness, number of the used data points (N) and its elevation standard deviation (SD). Due to the method difference we calculate the DEM uncertainty for observed and interpolated grid cells respectively. The surface slope and roughness are directly derived from the ICESat-2 DEM, the slope in one grid cell is derived as the maximum rate of change in elevation from that cell to its eight neighbors, the roughness is derived from the elevation difference between DEM and the smoothed DEM (by applying a 3 by 3 median filter). For observed grid cells, N is the number of the data points in each grid cell used for elevation estimation; for interpolated grid cells, N is derived by counting all data points within a search radius of 10 km, which is the radius used for elevation interpolation. SD is the standard deviation of elevations of these data points. The differences between DEM and OIB elevations are calculated and firstly binned w.r.t surface slope. The slope is divided into 200 bins with an interval of 0.01° (from 0 to 2°), the median and standard deviation are calculated for each bin. This processing method is also applied for other three parameters, an interval of 0.05 m for surface roughness, 250/500 (observed/interpolated grid cells) for N and 0.25 m for SD. For each distribution a 2-order polynomial is fitted by using the different standard deviations of the elevation differences for each bin. The corresponding coefficients are listed in Table 3. This kind of polynomial order ensure a good and robust fitting performance, including for the small elevation differences in flat regions. Finally, the DEM uncertainty is calculated as follows:

$$u = \sum_{i=1}^{4} w_i u_i$$

$$w_i = \frac{1}{s_i \sum_{i=1}^{4} \frac{1}{s_i}}$$

$$s_i = \frac{\sigma_i}{\sum_{i=1}^{4} \sigma_i}$$

$$u_i = b_{i1} x^2 + b_{i2} x + b_{i3}$$

Where $u$ is the DEM uncertainty, $w_i$ is the weighting factor and $u_i$ is the uncertainty for each uncertainty source. $s_i$ is the scaling factor and $\sigma$ is standard deviation of the difference between data and the polynomial fit. $b_{i\,0\text{-}3}$ are the coefficients for each polynomial fit (as listed in Table 3). When deriving the ICESat-2 DEM uncertainty estimation, the uncertainty from ICESat-2 measurements is not considered because the effect of ICESat-2 measurement bias is limited (< 5 cm, Brunt et al., 2019; < 14 cm, Shen et al., 2021).

Table 3. The fitting coefficients and weights used for the DEM uncertainty estimation

| Coefficient | Slope | Roughness | N | SD |
|---|---|---|---|---|

| | | | | | |
|---|---|---|---|---|---|
| Observed | $b_1$ | 0.13 | -0.02 | $-1.53\times10^{-9}$ | -0.01 |
| | $b_2$ | 6.20 | 0.90 | $-5.02\times10^{-5}$ | 0.42 |
| | $b_3$ | 3.37 | 4.37 | 12.13 | 4.85 |
| | Weights | 0.45 | 0.41 | 0.05 | 0.09 |
| Interpolated | $b_1$ | 0.38 | -0.02 | $2.96\times10^{-9}$ | $-4.98\times10^{-3}$ |
| | $b_2$ | 5.04 | 0.76 | $-3.60\times10^{-4}$ | 0.30 |
| | $b_3$ | 5.13 | 6.56 | 17.50 | 7.55 |
| | Weights | 0.49 | 0.37 | 0.06 | 0.08 |

Figure 7: I found some negative values in your DEM map in the boundary of ice shelves, can you explain them?

Negative elevation values are common for Ross ice shelf, these are also found in other DEMs, such as ICESat/ERS-1 DEM, Helm CryoSat-2 DEM, Slater CryoSat-2 DEM, REMA DEM and TanDEM PolarDEM (ICESat DEM does not have negative values, all its values $\geq 0$). Here we show the spatial distributions of negative elevation values in six DEMs (in black, as shown in the figure below), the extents and distributions are overall matched well.

One sentence about the negative values in DEM is also added in Section 3.1 in the revised text: '…*Negative elevations can be found in the ice shelves, especially in the Ross Ice Shelf.*'.

[Figure]

P13, L268: Can you prove more evidence here to clarify why ice sheet elevations are more accurate than those estimated for ice shelves.

In order to find the explanation, we present the histograms of surface slope and roughness values (derived from OIB data) for ice sheet and ice shelves in below:

[Figure]

**Figure.** Histograms of the OIB-derived surface slope and roughness values for ice sheet and ice shelves.

As we can found in this figure, observed ice shelves have overall smaller surface roughness than ice sheet, but have a larger percentage of high-slope areas than ice sheet. For example, approximately 70% of the OIB measurement points which covered ice sheet have slope values of < 0.01°. In comparison, approximately 50% of the OIB measurement points which located in ice shelves have slope values of < 0.01°. Hence, observed ice shelves have a higher percentage of high-slope areas, which may cause larger elevation biases. To test this argument, standardized regression coefficients between surface slope/roughness and the elevation difference (i.e., mean absolute difference between ICESat-2 DEM and OIB elevations) are calculated here by using a multivariate linear regression model (this model is fitted by using an iterative least-squares fit). All OIB data in 2018 and 2019 are used. Standardized values of surface slope, roughness and elevation difference are used for a valid comparison. The regression coefficients for surface slope and roughness are 0.18 and -0.01. Larger regression coefficient indicates that the surface slope has greater effect on elevation difference than roughness. Hence, although ice shelves observed by OIB data have smaller surface roughness than ice sheet, a higher percentage of high-slope areas makes ice shelves have a slight worse DEM performance. This discussion has also been mentioned in Section 3.2: '*Ice sheet elevations are more accurate than those estimated for ice shelves, **which may due to a higher percentage of high-slope areas in ice shelves observed by OIB data than in ice sheet**.*'.

Figure 9: I noticed that OIB elevations are near> 0 while your DEM has some elevations even less than -200 m, can you explain this?

Negative elevation values are common and these are also found in other DEMs, such as ICESat/ERS-1 DEM, Helm CryoSat-2 DEM, Slater CryoSat-2 DEM, REMA DEM and TanDEM PolarDEM (ICESat DEM does not have negative values, all its values ≥ 0). One sentence about the negative values in DEM is also added in Section 3.1 in the revised text: '*Negative elevations can be found in the ice shelves, especially in the Ross Ice Shelf.*'.

In the revised manuscript, to remove additional elevation outliers a 3-standard-deviation filter and a median filter were applied to the DEM, the generated DEM has been reevaluated based on the

OIB and GNSS data, this figure has also been changed and the results are more reasonable (without smaller negative elevation values).

[Figure]

Figure 9. Scatter plots of ICESat-2 DEM elevation and OIB airborne elevation (a) and GNSS elevation (b), respectively. The surface slopes are distinguished in different colours, as shown in the figure legend.

Table 5: why the number of used OIB measurement points in this table is different from that in Table 4.

Here OIB airborne elevation measurements in areas of low elevation change from 2009 to 2019 were used, while in Table 4 (now Table 5 in the revised text) OIB airborne elevation measurements in areas of low elevation change from 2009 to 2017 and in the whole Antarctica from 2018 to 2019 were used.

Table 5: here you compared the other DEM and your DEM to OIB data, as the same OIB data were used, but the timestamps of DEMs are different, do you consider this effect or how to reduce it?

Yes, we have considered this effect as listed in Section 2.2 in the previous manuscript, OIB and GNSS data in the low elevation-change areas and OIB data with small time-difference (< one year) comparing to ICESat-2 DEM are used for DEM evaluation. However, the effect of time difference between the DEM and evaluation data still needs to be considered. In the revised manuscript, we adjust the changes of ICESat-2 DEM elevation values which occur during the time difference between these two data by using the trend values derived from Smith et al. (2020) and we assume the constant elevation change rates, the corresponding adjustments are then calculated and applied for ICESat-2 DEM in the locations of OIB/GNSS measurements. The related text, figures and tables have been revised, the same conclusions are derived (as shown in the very beginning).

This part has been added in Section 2.2 in the revised text: '*Although OIB and GNSS data in the low elevation-change areas and OIB data with small time-difference (< one year) comparing to ICESat-2 DEM are used for DEM evaluation, the effect of time difference between the DEM and evaluation data still needs to be considered. Here, we adjust the changes of ICESat-2 DEM elevation values which occur during the time difference between these two data, the trend values are derived from Smith et al. (2020) and we assume the constant elevation change rates, the corresponding*

*adjustments are calculated and applied for the DEM values in the locations of OIB/GNSS measurements before comparisons.*'.

P19, L385: Again, please provide an elevation change rate map to evaluate the elevation estimation performance.

This has been responded in the comment above.

Please make a revision of the manuscript accordingly, I recommend this manuscript be considered as an ESSD publication after a revision.

Thank you for your recommendation, we have revised the manuscript accordingly based on the comments above.

Regards,

---

## Author Comment (AC2)

We thank the reviewer for the helpful feedback, these suggestions have significantly improved the data and text, we are appreciative of the help and time.

We have addressed all the comments here, point by point responses to the comments are listed in BLUE.

Here we summarize the major revision in the revised manuscript:

According to the reviewers, additional filters (a 3-standard-deviation filter and a median filter) have been applied in the generated DEM to improve the DEM performance. The new DEM was also evaluated by comparing to the OIB and GNSS data, similar performances can be found (as listed in the Tables below). In particular, to account for the temporal difference between the DEM and OIB/GNSS data, when performing the comparisons we adjusted the ICESat-2 DEM elevations for the surface elevation changes between the acquisition periods of these two data. The adjustments were calculated by using the trend values derived from Smith et al. (2020) and we assumed the constant elevation change rates, these were applied for the DEM values in the locations of OIB/GNSS measurements. The related text, figures and tables have been accordingly revised, the same conclusions are derived.

The updated DEM map (including uncertainty map) can be downloaded from Google drive at https://drive.google.com/drive/folders/1h0-QxAkjGMSc-eqlBBigiqvgp0k-8iIB?usp=sharing at this stage, and we will replace the previous revision in the data storage (i.e., National Tibetan Plateau Data Center, Institute of Tibetan Plateau Research, Chinese Academy of Sciences) after the manuscript revision.

Table 4 (previous Table 3). Comparisons between the ICESat-2 DEM and OIB airborne elevation measurements (including data in areas of low elevation change from 2009 to 2017 and data in the Antarctica from 2018 to 2019) in observed and interpolated areas for individual regions (i.e., the ice sheet and ice shelves). MeD: median deviation, MeAD: median absolute deviation, SD: standard deviation, RMSD: root-mean-square deviation.

| | Region | MeD (m) | MeAD (m) | SD (m) | RMSD (m) | Number of used OIB measurement points |
|---|---|---|---|---|---|---|
| Observed | Ice sheet | -0.17 | 1.21 | 9.25 | 9.26 | 3589087 |
| | Ice shelves | 0.59 | 2.53 | 14.07 | 14.09 | 191754 |
| | Total | -0.15 | 1.26 | 9.56 | 9.57 | 3780841 |
| Interpolated | Ice sheet | -0.52 | 2.63 | 13.30 | 13.36 | 1237416 |
| | Ice shelves | 0.44 | 3.00 | 15.16 | 15.21 | 185613 |
| | Total | -0.41 | 2.67 | 13.58 | 13.62 | 1423029 |
| Overall | Ice sheet | -0.22 | 1.47 | 10.44 | 10.47 | 4826503 |
| | Ice shelves | 0.53 | 2.75 | 14.62 | 14.65 | 377367 |
| | Total | -0.19 | 1.54 | 10.81 | 10.83 | 5203870 |

Table 7 (previous Table 6). Comparisons between the ICESat-2 DEM, ICESat DEM, ICESat/ERS-1 DEM, Helm CryoSat-2 DEM, Slater CryoSat-2 DEM, REMA DEM, TanDEM PolarDEM and

| GNSS elevation data in areas of low elevation change from 2001 to 2015. | | | | | |
|---|---|---|---|---|---|
| | MeD (m) | MeAD (m) | SD (m) | RMSD (m) | Number of used GNSS measurement points |
| ICESat-2 DEM | 0.02 | 0.50 | 1.59 | 1.60 | |
| ICESat DEM | -3.79 | 4.30 | 10.99 | 13.10 | |
| ICESat/ERS-1 DEM | -0.75 | 1.02 | 2.22 | 2.32 | |
| Helm CryoSat-2 DEM | 0.16 | 0.89 | 2.84 | 2.92 | 488963 |
| Slater CryoSat-2 DEM | -0.12 | 0.61 | 2.41 | 2.43 | |
| REMA DEM | 0.06 | 0.30 | 0.78 | 0.78 | |
| TanDEM PolarDEM | -4.03 | 4.03 | 1.52 | 4.34 | |

Review:

A new digital elevation model of Antarctica derived from ICESat-2

Shen et.al. 2022

The study presents a new elevation model of Antarctica based on one-year of ICESat-2 observations. The authors provide a specific time-stamped DEM with a final pixel size of 500m, following the same approach as presented by Slater et.al. (2017).

The new DEM is validated against OIB and GNSS data and compared to existing Antarctic DEMs. Results show an improved accuracy compared to DEMs based on Radar altimetry but with less accuracy than DEMs based on Radar interferometry or Stereo-Photogrammetry.

In general, it is an interesting project and worth to be published as ICESat2 provides precise point information with high accuracy and good coverage. This large data base should be used to generate a gridded data product of high quality which is easily accessible and to be used in different applications. The authors did this approach in a comprehensible way. The paper was already submitted to TC and underwent a review process. Most of points of my former review were answered and corrected.

The paper is well structured, methods are explained and figures are of high quality. The validation against OIB and GNSS is clear and shows at least in numbers an improved DEM compared to other Altimeter based DEMs.

Data is accesible via the data link

Thank you very much for your positive feedback and advice.

Based on the paper I was now curious to have a look to the DEM itself.
However, after looking at the data I'm a bit disappointed and have some question marks with respect of data quality and usability of the dataset. Attached are two screenshots of the DEM underlain by its hillshade. It can be seen that there are a lots of artefacts visible, even over the flat lake Vostok. Elevation differences at those erroneous pixels are in the order of meters to tenth of meters. In addition a grid like structure is visible in the hillshade or in a roughness image created from the

DEM itself. Furthermore the uncertainty map makes absolutely no sense to me. One can see tracks with uncertainties of 0.01m and between those tracks the values jump to 30 / 50m or more.

I'm wondering why the erroneous pixels are not seen in the statistics of the validation. I think the authors should re-think their methodology in respect to outlier detection as well as the uncertainty. Where are the outliers come from? Why do you have such large jumps in the uncertainty map.

Based on the data set itself I cannot recommend a publication at the current stage as to my opinion this data set is not useful because there are too many erroneous pixels all over Antarctica.

1) about the artefacts
In the previous manuscript, ICESat-2 data in bad quality (based on the data quality flag) are not used and then the estimated elevations due to the poor fitting performances in the grid cells (i.e., Table 2) are removed to ensure the generated DEM quality. Nevertheless, some artefacts still occurred in the DEM map, and these values were derived from the model fitting method (not the interpolation). That is to say, although some quality control criteria have been applied, noises or artefacts cannot be totally removed. In the revised manuscript, to remove additional elevation outliers a 3-standard-deviation filter is applied to the DEM. Visual inspection indicates that a large number of artefacts are removed and the remaining is further removed by using a median filter. After the application of these filters, the ICESat-2 DEM shows a reasonable and acceptable performance. The corresponding hillshade maps in some regions are shown below. We also evaluated the new DEM by comparing to the OIB and GNSS data, similar performances can be found (as listed in the Tables above), this means that the amount of the artefacts in previous DEM is relatively small, and thus has limited effect on the evaluation result.

The related statements have been added into the revised manuscript (see subsection 2.4.2):
'… *Finally, in order to remove additional elevation outliers in the generated DEM, a 3-standard-deviation filter (3 by 3) is firstly applied. Visual inspection indicates that only a small number of anomalous elevations remain and these are further removed by using a 3 by 3 median filter. These quality assurance filters ensure the elevation pattern of the final DEM is smoothed and reasonable.*'.

[Figure]

[Figure]

[Figure]

[Figure]

Figure. Shaded relief maps in some regions derived from the ICESat-2 DEM.

2) about the abnormal structure

We have checked the whole generation processes and found a mistake when merging three DEMs (500m DEM, 1000m DEM and interpolated DEM), this caused a slight spatially offset between different DEMs and thus caused the grid like structures in the hillshade map. In the revised manuscript this has been corrected and the corresponding hillshade map is reasonable now (as shown in below).

[Figure]

Figure. One example of the shaded relief maps in Antarctica derived from the ICESat-2 DEM.

3) about the uncertainty map

Here DEM uncertainties are calculated for observed and interpolated grid cells, respectively. The observed grid cell uncertainty is derived based on the model fitting performance and the interpolated grid cell uncertainty is calculated from the kriging variance error. As a series of quality control criteria have been applied to remove the unrealistic elevations due to the poor fitting performances in the observed grid cells, the uncertainty values are thus relatively small. While for interpolated grid cells, the elevations are derived based on the kriging interpolation, no valid ICESat-2 measurement points are included in these grid cells, hence the uncertainty values are usually large. We have compared the uncertainty values in the interpolated grid cells to these in CryoSat-2 DEM

(Slater et al., 2018) which also derived from kriging interpolation and also found the similar uncertainty values, this means that our uncertainty estimation method is right.

As our DEM is generated by combining the 500m DEM, 1000m DEM and interpolated DEM, between the tracks the elevations are derived from the interpolated DEM, larger uncertainty values can be found comparing to those tracks (elevations are derived from model fitting method) with small uncertainties. Due to the method difference between the model fitting method and interpolation method, especially in the interpolated grid cells no valid ICESat-2 points can be used, it is natural that their uncertainties have some differences.

Here we also derive an uncertainty map based on the approach in Helm et al. (2014). The OIB elevation data are used as the reference and the elevation differences due to the time difference between OIB data and DEM are corrected based on the elevation-change rates from Smith et al. (2020). The DEM uncertainty is then calculated from surface slope, roughness, number of the used data points (N) and its elevation standard deviation (SD). Due to the method difference we calculate the DEM uncertainty for observed and interpolated grid cells respectively. The surface slope and roughness are directly derived from the ICESat-2 DEM, the slope in one grid cell is derived as the maximum rate of change in elevation from that cell to its eight neighbors, the roughness is derived from the elevation difference between DEM and the smoothed DEM (by applying a 3 by 3 median filter). For observed grid cells, N is the number of the data points in each grid cell used for elevation estimation; for interpolated grid cells, N is derived by counting all data points within a search radius of 10 km, which is the radius used for elevation interpolation. SD is the standard deviation of elevations of these data points. The differences between DEM and OIB elevations are calculated and firstly binned w.r.t surface slope. The slope is divided into 200 bins with an interval of 0.01° (from 0 to 2°), the median and standard deviation are calculated for each bin. This processing method is also applied for other three parameters, an interval of 0.05 m for surface roughness, 250/500 (observed/interpolated grid cells) for N and 0.25 m for SD. For each distribution a 2-order polynomial is fitted by using the different standard deviations of the elevation differences for each bin. The corresponding coefficients are listed in Table 3. This kind of polynomial order ensure a good and robust fitting performance, including for the small elevation differences in flat regions. Finally, the DEM uncertainty is calculated as follows:

$$u = \sum_{i=1}^{4} w_i u_i$$

$$w_i = \frac{1}{s_i \sum_{i=1}^{4} \frac{1}{s_i}}$$

$$s_i = \frac{\sigma_i}{\sum_{i=1}^{4} \sigma_i}$$

$$u_i = b_{i1} x^2 + b_{i2} x + b_{i3}$$

Where $u$ is the DEM uncertainty, $w_i$ is the weighting factor and $u_i$ is the uncertainty for each uncertainty source. $s_i$ is the scaling factor and $\sigma$ is standard deviation of the difference between data and the polynomial fit. $b_{i\,0-3}$ are the coefficients for each polynomial fit (as listed in below).

Table 3. The fitting coefficients and weights used for the DEM uncertainty estimation

|  | Coefficient | Slope | Roughness | N | SD |
|---|---|---|---|---|---|
| Observed | $b_1$ | 0.13 | -0.02 | $-1.53 \times 10^{-9}$ | -0.01 |
|  | $b_2$ | 6.20 | 0.90 | $-5.02 \times 10^{-5}$ | 0.42 |
|  | $b_3$ | 3.37 | 4.37 | 12.13 | 4.85 |
|  | Weights | 0.45 | 0.41 | 0.05 | 0.09 |
| Interpolated | $b_1$ | 0.38 | -0.02 | $2.96 \times 10^{-9}$ | $-4.98 \times 10^{-3}$ |
|  | $b_2$ | 5.04 | 0.76 | $-3.60 \times 10^{-4}$ | 0.30 |
|  | $b_3$ | 5.13 | 6.56 | 17.50 | 7.55 |
|  | Weights | 0.49 | 0.37 | 0.06 | 0.08 |

Additionally, by comparing to the OIB or GNSS elevation data, we can estimate the actual ICESat-2 DEM uncertainty as the SD of the differences to OIB or GNSS elevation data. In this estimated uncertainty map (Fig. 5b, as shown in below), a median value of 5.84 ± 5.29 m can be found. The SD of differences to OIB data which obtained in the large scale shows a value of 10.44 m (Table 4, including plenty of measurements in ice sheet margin), while in the ice sheet interior a value of 3.26 m is found (Table 6). Considering the data coverage and surface-slope difference, the estimated uncertainty values can represent the SDs from what is given as OIB, which means that the provided uncertainty estimates are reliable. Small SD value of 1.59 m can be found when comparing to the GNSS data (Table 7) which were obtained in the regions of low slope, this may due to the resolution and measurement accuracy differences between airborne and GNSS data, hence the ICESat-2 DEM uncertainty map may be slightly overestimated and can be assumed as the upper limit.

Slight jumps (< 3 m) can also be found in the current uncertainty map, this is due to the method difference when deriving the elevations and this pattern is consistent to this of elevation source. Due to the method difference between model fitting and interpolation, it is natural that different uncertainty values are found. We think the current uncertainty map (as shown in below) provides more reasonable elevation uncertainties than previous one and use it in the revised manuscript.

[Figure]

[Figure]

Figure 5. (a) A new DEM of Antarctica at a posting of 500 m derived from ICESat-2, which covers both the ice sheet and ice shelves with the southern limit of 88°S. (b) Map of the ICESat-2 DEM elevation uncertainty.

---

## Author Comment (AC3)

We thank the reviewer for the feedback, we have addressed all the comments here, point by point responses to the comments are listed in BLUE.

This manuscript generates a 500 m resolution DEM of Antarctica based on the ICESat-2 data from November 2018 to November 2019 using a spatio-temporal fitting approach. The authors validated the DEM using IceBridge airborne altimetry data and GNSS ground measurements, and also compared it with six other published Antarctic DEMs. Although the results show that the accuracy of this DEM is very superior, I am doubtful about this result.

General Comments:

It is a good attempt to build an Antarctic DEM dataset based on ICESat-2 altimetry data, and Antarctic DEM is important for the study of Antarctic ice sheet changes. ICESat-2 satellite can indeed provide very high resolution and high accuracy ice sheet elevation data and has great potential to be a reliable data source for building Antarctic DEM. However, unfortunately, the dataset was not completely utilized by this manuscript.

We thank the reviewer for the helpful suggestions, we have revised the manuscript accordingly and the details are listed below. Here we summarize the major revision in the revised manuscript:

According to the reviewers, additional filters (a 3-standard-deviation filter and a median filter) have been applied in the generated DEM to improve the DEM performance. The new DEM was also evaluated by comparing to the OIB and GNSS data, similar performances can be found (as listed in the Tables below). In particular, to account for the temporal difference between the DEM and OIB/GNSS data, when performing the comparisons we adjusted the ICESat-2 DEM elevations for the surface elevation changes between the acquisition periods of these two data. The adjustments were calculated by using the trend values derived from Smith et al. (2020) and we assumed the constant elevation change rates, these were applied for the DEM values in the locations of OIB/GNSS measurements. The related text, figures and tables have been accordingly revised, the same conclusions are derived.

The updated DEM map (including uncertainty map) can be downloaded from Google drive at https://drive.google.com/drive/folders/1h0-QxAkjGMSc-eqlBBigiqvgp0k-8iIB?usp=sharing at this stage, and we will replace the previous revision in the data storage (i.e., National Tibetan Plateau Data Center, Institute of Tibetan Plateau Research, Chinese Academy of Sciences) after the manuscript revision.

Table 4 (previous Table 3). Comparisons between the ICESat-2 DEM and OIB airborne elevation measurements (including data in areas of low elevation change from 2009 to 2017 and data in the Antarctica from 2018 to 2019) in observed and interpolated areas for individual regions (i.e., the ice sheet and ice shelves). MeD: median deviation, MeAD: median absolute deviation, SD: standard deviation, RMSD: root-mean-square deviation.

| | Region | MeD (m) | MeAD (m) | SD (m) | RMSD (m) | Number of used OIB measurement points |
|---|---|---|---|---|---|---|
| Observed | Ice sheet | -0.17 | 1.21 | 9.25 | 9.26 | 3589087 |

|  |  |  |  |  |  |
|---|---|---|---|---|---|
|  | Ice shelves | 0.59 | 2.53 | 14.07 | 14.09 | 191754 |
|  | Total | -0.15 | 1.26 | 9.56 | 9.57 | 3780841 |
| Interpolated | Ice sheet | -0.52 | 2.63 | 13.30 | 13.36 | 1237416 |
|  | Ice shelves | 0.44 | 3.00 | 15.16 | 15.21 | 185613 |
|  | Total | -0.41 | 2.67 | 13.58 | 13.62 | 1423029 |
| Overall | Ice sheet | -0.22 | 1.47 | 10.44 | 10.47 | 4826503 |
|  | Ice shelves | 0.53 | 2.75 | 14.62 | 14.65 | 377367 |
|  | Total | -0.19 | 1.54 | 10.81 | 10.83 | 5203870 |

Table 7 (previous Table 6). Comparisons between the ICESat-2 DEM, ICESat DEM, ICESat/ERS-1 DEM, Helm CryoSat-2 DEM, Slater CryoSat-2 DEM, REMA DEM, TanDEM PolarDEM and GNSS elevation data in areas of low elevation change from 2001 to 2015.

| | MeD (m) | MeAD (m) | SD (m) | RMSD (m) | Number of used GNSS measurement points |
|---|---|---|---|---|---|
| ICESat-2 DEM | 0.02 | 0.50 | 1.59 | 1.60 | |
| ICESat DEM | -3.79 | 4.30 | 10.99 | 13.10 | |
| ICESat/ERS-1 DEM | -0.75 | 1.02 | 2.22 | 2.32 | |
| Helm CryoSat-2 DEM | 0.16 | 0.89 | 2.84 | 2.92 | 488963 |
| Slater CryoSat-2 DEM | -0.12 | 0.61 | 2.41 | 2.43 | |
| REMA DEM | 0.06 | 0.30 | 0.78 | 0.78 | |
| TanDEM PolarDEM | -4.03 | 4.03 | 1.52 | 4.34 | |

Other specific comments

First of all, there is no innovation in the method study as the spatio-temporal method was referred from Slater et al. (2018). On the one hand, although Slater et al. used this method to build a superior performance Antarctic DEM based on data from the radar altimetry satellite CryoSat-2, the authors' transposition of this method to the data processing of the laser altimetry satellite ICESat-2 may create unknown uncertainties. On the other hand, I consider that using the altimetry data with only a time span of 1 year cannot show the priority of the spatio-temporal fitting model, and can cause fitting errors due to the limited data density and spatial distribution. In fact, this is also reflected in the manuscript, where only 46% of the grids in the 500m resolution DEM claimed by the authors are directly generated by fitting sampling points within the 500m grids, with other gaps either obtained by resampling the grids at low resolution or by kriging interpolation.

Both the elevation measurement data are obtained from the CryoSat-2 radar altimeter and ICESat-2 laser altimeter, hence the kind of input data for spatio-temporal fitting method in our study and Slater et al. (2018) are the same. Additionally, as the CryoSat-2 radar signals may penetrate the snow layer, the elevation measurements from ICESat-2 tend to have less uncertainty than those from CryoSat-2 and ICESat-2 is thus expected to have better performance. Hence, we think that the spatio-temporal method can also be used for the laser altimeter data.

We agree that the spatio-temporal fitting method may be more appropriate for longer time series of altimeter data. However, if this method can still separate temporal elevation changes with just one year of data, it still can be used for DEM generation. Hereafter we provide the map of elevation

change rate estimated from this study, we also provide the estimation result from 2003 to 2019 in Smith et al. (2020) for a comparison. Overall, considering the time difference similar elevation change patterns can be found between the two figures. For example, larger elevation decreases can be found in the margin of West Antarctica, obvious elevation increases can be found in the interior of West Antarctica (red cycles in the figure). The elevation change pattern based on one-year ICESat-2 data is reasonable, which indicates that one year of data can give a reliable elevation change map and the elevation estimation is thus reliable. This may due to the much higher measurements density and accuracy of ICESat-2 than previous altimeters.

[Figure]

**Figure.** Map of elevation change rate in Antarctica derived from one year of ICESat-2 data in this study (left) and map of elevation change rate in Antarctica from 2003 to 2019 in Smith et al. (2020) (right).

One-year of satellite altimeter data have also been used to generate the DEM for Antarctica in previous studies. In Helm et al. (2014), one-year of CryoSat-2 data were used for the DEM generation. The DEM is in the spatial resolution of 1 km and the gaps were filled by using the ordinary kriging interpolation (also a series of processing scheme was included), a quite good performance can be found when comparing to the ICESat elevation data. ICESat-2 has much denser and larger coverage than CryoSat-2, hence it is also reasonable to derive the Antarctic DEM by using one-year of ICESat-2 data considering the performance of one-year CryoSat-2 data. In addition, approximately $4.69 \times 10^9$ ICESat-2 measurement points from November 2018 to November 2019 were used in this study, while $2.5 \times 10^8$ CryoSat-2 measurement points from July 2010 to July 2016 were used in Salter et al. (2018). Considering this, although only one year of ICESat-2 data were used the data density is still larger than seven years of CryoSat-2 data.

The resolutions of grid cells (i.e., 500 m and 1 km) are appropriate for the used ICESat-2 data, it is possible for a quadratic form to model the topography at these scales and smaller elevation residuals can be found than using a simple linear fit (Flament and Remy, 2012). Due to the coverage of used ICESat-2 data (one year of data), we firstly generate the DEMs in 500 m and 1 km grid cells and approximately 74% of Antarctica can be covered, the remaining observation gaps are interpolated

using the ordinary kriging method; in Slater et al. (2018) a 60% coverage can be found for his finest 1km grid DEM by using seven years of CryoSat-2 data. For the 1 km grid DEMs from this study and Slater et al. (2018), although the time series are different similar data coverage can be found.

In addition, ICESat-2 DEM has a comparable performance to other DEMs by comparing to the same airborne and GNSS data sets (after the correction for effect of the temporal difference between the DEM and OIB/GNSS data), which also proves the feasibility of the data and method.

Considering the data density/coverage, method performance and DEM accuracy, the DEM generated by using spatio-temporal fitting method from one-year of ICESat-2 data is still reasonable and reliable.

The related statements about the reliability of using one-year of ICESat-2 data have also been added into the revised manuscript (i.e., subsection 2.4.1):
'... *Additionally, the performance of the surface fit method also depends on the timespan of the input data, that is to say, it should be noted that whether one-year of ICESat-2 data can be used to obtain a satisfied fitting performance. Here we find that the elevation-change rate map based on one-year ICESat-2 data (i.e., $a_5$ in Eq.1) has a similar pattern with that from Smith et al. (2020), which estimated the elevation-change rate from 2003 to 2019 based on ICESat and ICESat-2 data, indicating that one-year of data can also provide the reasonable elevation change rates and thus the surface fit method used here is reliable.*'

The related statements about the choice of DEM resolution and details about the spatial coverage have also been listed in subsection 2.4.2 in the manuscript.

References:
Smith B, Fricker H A, Gardner A S, et al. Pervasive ice sheet mass loss reflects competing ocean and atmosphere processes. Science, 2020, 368(6496): 1239-1242.
Slater T, Shepherd A, McMillan M, et al. A new digital elevation model of Antarctica derived from CryoSat-2 altimetry. The Cryosphere, 2018, 12(4): 1551-1562.
Helm, V., Humbert, A., and Miller, H.: Elevation and elevation change of Greenland and Antarctica derived from CryoSat-2, The Cryosphere, 8, 1539-1559, https://doi.org/ 10.5194/tc-8-1539-2014, 2014.

Secondly, I have serious doubts about the reliability of this Antarctic DEM dataset, and although the authors use some measured data to validate it, I do not think this validation method is reliable. Although these variations are neglected in areas with small elevation changes in the interior of the ice cap, I do not agree that it is reasonable to use OIB and GNSS data with large time differences to assess the DEM accuracy. In addition, it is not representative of the accuracy of the whole DEM, limited by the amount and distribution of the validation data. Of course, this is due to the limitation of obtaining large-scale field measurement data. However, it cannot be arbitrarily claimed that the accuracy of DEM under such validation conditions is better than the results of other scholars.

In order to provide a relatively comprehensive and robust evaluation of ICESat-2 DEM, all OIB and GNSS data in areas of low elevation change are used. The CryoSay-2 Low Rate Mode (which was designed for flat ice sheet interior measurements) mask is used to extract the regions of low elevation change. CryoSat Geographical Mode Mask (v 4.0, updated in 19 August - 26 August 2019) at https://earth.esa.int/eogateway/news/cryosat-geographical-mode-mask-4-0-released is used here.

One problem may be caused when using OIB and GNSS data with time differences to assess the DEM accuracy, that is the 'real' elevations may be variable during the timespan. As we can find in areas of low elevation change, the elevation change rate is about **-0.0074±0.0821 m/yr** from 2003 to 2019, in these areas the effect of the elevation change on the DEM evaluation can be ignored. However, we do agree that the effect of the time difference between OIB/GNSS data and ICESat-2 DEM should be considered. In the revised manuscript, we adjust the ICESat-2 DEM elevations for the surface elevation changes between the acquisition periods of these two data and we assume the constant elevation change rates. The adjustments are calculated by using the trend values are derived from Smith et al. (2020) and applied for the DEM values in the locations of OIB/GNSS measurements. The related text, figures and tables have been revised, but the same conclusions are derived (according to the Tables in the very beginning).

This part has also been added in Section 2.2 in the revised text: '*Although OIB and GNSS data in the low elevation-change areas and OIB data with small time-difference (< one year) comparing to ICESat-2 DEM are used for DEM evaluation, the effect of time difference between the DEM and evaluation data still needs to be considered. Here, we adjust the changes of ICESat-2 DEM elevation values which occur during the time difference between these two data, the trend values are derived from Smith et al. (2020) and we assume the constant elevation change rates, the corresponding adjustments are calculated and applied for the DEM values in the locations of OIB/GNSS measurements before comparisons.*'.

Besides, in the revised manuscript we mainly focus on the evaluation of ICESat-2 DEM, construct a general comparison to other DEMs and avoid the expression about the accuracy rank. In Section 4 in the revised text, we also add a paragraph to emphasize it:
'*… It should be noted that, the spatio-temporal coverages of used OIB and GNSS data are limited here, and they cannot provide an unbiased evaluation for ICESat-2 DEM and other DEMs. Hence the comparisons above only give a general reference for their performances and cannot be used as the quantitative accuracy evaluation.*'.

In addition, from the perspective of manuscript writing, this manuscript is well structured and the language is more fluent, but there are some places where the expression is not very clear and there are also a large number of obvious typographical errors. For example, Fig. 1b and Fig. 1c are not seen in Fig. 1, but they appear in lines 105 and 108, respectively; in line 305, the description of the comparison of other DEMs should be discussed in Section 4, which seems very confusing here; in Table 6. it should be 'Number of used GNSS measurement points' instead of 'Number of used OIB measurement points', etc.

We have carefully checked the whole manuscript, corrected the related errors and revised the vague expressions.

For Fig. 10 (line 305 in previous manuscript) we only focus on the evaluation of ICESat-2 DEM now: '*In the ice sheet interior where surface slopes are small (Fig. 10a), elevation differences of approximately 5 m can be found (**the median elevation differences for ICESat-2 DEM is -0.13±0.19 m**). The elevation differences are further reduced when surface slope become smaller. While at the Pine Island Glacier where surface slopes are large (Fig. 10b), elevation differences of approximately 20 m can be found in the undulated terrains (**the median elevation differences for ICESat-2 DEM is -0.01±4.58 m**). Overall, ICESat-2 DEM has better performances in the flat regions than steep areas. Regions of low surface slope represent the majority of Antarctic ice sheet, hence most elevations from ICESat-2 DEM have smaller elevation biases.*'. The description of the comparisons of other DEMs has been removed.

In fact, I have also seen this manuscript in the discussion forum of *The Cryosphere* last year, and this does not seem to be too much changed from the previous manuscript. Moreover, another article by the authors using the same approach applied to Greenland has been published in *ESSD* (Fan et al. 2022), and the two manuscripts are similar in approach and writing style, and I do not think it is worth publishing a similar work again.

In conclusion, I think the manuscript has no innovations in the DEM generation method and the dataset is not reliable, and its validation data are not enough to support the authors' conclusion, so it is not recommended for publication.

References
Fan, Y., Ke, C.-Q., and Shen, X.: A new Greenland digital elevation model derived from ICESat-2 during 2018–2019, Earth Syst. Sci. Data, 14, 781–794, https://doi.org/10.5194/essd-14-781-2022, 2022.
Slater, T., Shepherd, A., McMillan, M., Muir, A., Gilbert, L., Hogg, A. E., Konrad, H., and Parrinello, T.: A new digital elevation model of Antarctica derived from CryoSat-2 altimetry, The Cryosphere, 12, 1551–1562, https://doi.org/10.5194/tc-12-1551-2018, 2018.

We made a major revision for the manuscript according to the comments from three reviewers of *The Cryosphere*, we have regenerated and reevaluated the DEM by comparing to the OIB/GNSS data. The presented DEM and evaluation results have been greatly changed, the manuscript structure is the same and hence it seems like that there are no too much changes.

The same method in Slater et al. (2018) is used here to generate the DEM from ICESat-2 data, in the comments above we have provided the reasons why we choose this method and proved that one-year of data can still be used for DEM generation. More importantly, our result demonstrates that the ICESat-2 DEM can be provided in a sustainable way, i.e., the ICESat-2 DEM can be updated annually and thus accumulated on an annual base, which has large application potential for Antarctica especially under the warm climate. Additionally, reasonable elevation-change rate can also be obtained when deriving this DEM, which can provide an additional reference for ice topography and mass balance estimation. Hence, in Section 4 we add a paragraph to point out the

differentiation between our work and other studies, i.e., the special contribution of our work:

'... *Comparing to other DEMs, elevation change rate can be obtained when deriving the ICESat-2 DEM, which provides an additional reference for ice topography and mass balance estimation. Additionally, in previous studies several years of altimeter data are needed to derive the DEM in Antarctica. Due to the high-density measurements of ICESat-2, 13 months of ICESat-2 data can be used to generate a DEM for Antarctica and the performance is comparable to other DEMs, indicating that the ICESat-2 DEM can be updated annually. This study demonstrates the feasibility and reliability of using one-year ICESat-2 data to derive the Antarctic DEM, provides a reference for the processing scheme of DEM (e.g., in higher resolution, regularly updated) based on ICESat-2 in future.*'.

Although the structure of this paper has some similarities to Fan et al. (2022), it still has many differences, e.g., the comparison and choice of DEM resolutions, DEM postprocessing method, DEM uncertainty estimation method, DEM evaluation method (including slope-related and along-track comparisons), presentment of detailed maps of DEM, potential applications and advantages of this dataset, these can provide a potential reference for the generated DEM in future studies.

Our paper is a kind of 'Data description paper', its aim is to introduce the presented Antarctic DEM from ICESat-2 data, hence we think the dataset itself and the significance of this dataset/work to the scientific community are more important here. The related validation approach and statements have been changed/revised accordingly to provide reasonable evaluation and expressions. Similar evaluation result can also be found for our ICESat-2 DEM when using the new validation approaches (as shown in the above comments), hence this DEM is still reliable. More importantly, as you mentioned in the very beginning, Antarctic DEM is essential for the study of Antarctic ice sheet changes. The elevations of high accuracy and ability of annual update make the ICESat-2 DEM a special addition to the existing Antarctic DEM groups, and it can be further used for other scientific applications in Antarctica, and thus we still think that this paper still suits the scope of *ESSD*.